



# Retrieval of atmospheric $CH_4$ vertical information from TCCON FTIR spectra

Minqiang Zhou[1], Bavo Langerock[1], Mahesh Kumar Sha[1], Nicolas Kumps[1], Christian Hermans[1], Christof Petri[2], Thorsten Warneke[2], Huilin Chen[3], Jean-Marc Metzger[4], Rigel Kivi[5], Pauli Heikkinen[5], Michel Ramonet[6], and Martine De Mazière[1]

[1]Royal Belgian Institute for Space Aeronomy (BIRA-IASB), Brussels, Belgium
[2]Institute of Environmental Physics, University of Bremen, Bremen, Germany
[3]Centre for Isotope Research (CIO), Energy and Sustainability Research Institute Groningen (ESRIG), University of Groningen, Groningen, the Netherlands
[4]UMS 3365 – OSU Réunion, Université de La Réunion, Saint-Denis, Réunion, France
[5]Finnish Meteorological Institute, Space and Earth Observation Centre, Sodankylä, Finland
[6]Laboratoire des Sciences du Climat et de l'Environnement (LSCE/IPSL), UMR CEA-CNRS-UVSQ, Gif-sur-Yvette, France

**Correspondence:** Minqiang Zhou (minqiang.zhou@aeronomie.be)

**Abstract.** TCCON (Total Carbon Column Observing Network) column-averaged dry air mole fraction of $CH_4$ ($X_{CH_4}$) measurements have been widely used to validate satellite observations and to estimate model simulations. The GGG2014 code is the standard TCCON retrieval software performing a profile scaling retrieval. In order to obtain several vertical information in addition to total column, in this study, the SFIT4 retrieval code is applied to retrieve $CH_4$ mole fraction vertical profile using TCCON spectra (SFIT4TCCON) at six sites (Ny-Ålesund, Sodankylä, Bialystok, Bremen, Orléans and St Denis) during the time period of 2016 – 2017. The retrieval strategy of SFIT4TCCON is investigated. The degree of freedom for signal of the SFIT4TCCON retrieval is about 2.4, with two distinct species of information in the troposphere and in the stratosphere. The averaging kernel and error budget of the SFIT4TCCON retrieval are presented. The data accuracy and precision of the SFIT4TCCON retrievals, including the total column and two partial columns (in the troposphere and stratosphere), are estimated by TCCON standard retrievals, ground-based in situ measurements, ACE-FTS satellite observations, TCCON proxy data and AirCore measurements. By comparison against TCCON standard retrievals, it is found that the retrieval uncertainty of SFIT4TCCON $X_{CH_4}$ is similar to that of TCCON standard retrievals with the systematic uncertainty within 0.35% and the random uncertainty about 0.5%. The tropospheric and stratospheric $X_{CH_4}$ from SFIT4TCCON retrievals are assessed by comparing with AirCore measurements at Sodankylä, and there is a 1.2% overestimation in the SFIT4TCCON tropospheric $X_{CH_4}$ and a 4.0% underestimation in the SFIT4TCCON stratospheric $X_{CH_4}$, which are within the systematic uncertainties of SFIT4TCCON retrieved partial columns in the troposphere and stratosphere, respectively.

## 1 Introduction

The Total Carbon Column Observing Network (TCCON) is an international network established in 2004 using ground-based Fourier transform infrared (FTIR) spectrometer to record direct solar absorption spectra in the near-infrared (NIR) spectral



range and to retrieve from these spectra total columns of atmospheric greenhouse gases, including methane ($CH_4$) (Wunch et al., 2011). Currently, there are about 25 TCCON sites around the world with a latitude coverage from 45°S to 80°N. The standard TCCON retrieval code is GGG2014 (developed and maintained at JPL, NASA); it performs a profile scaling retrieval. TCCON provides the dry air total column averaged mole fraction of $CH_4$ ($X_{CH_4}$), which have been compared to and indirectly calibrated by the Infrastructure for the Measurement of the Europe Carbon Cycle (IMECC) profiles over the European TCCON stations, the high-performance instrumented airborne platform for environmental research (HIAPER) Pole-to-Pole Observations (HIPPO) profiles over the TCCON stations in Northern America East Asia and Oceania, and several AirCore profiles (Karion et al., 2010) over Lamont (USA). A scaling factor of 0.977±0.002 ($1\sigma$) is applied to the retrieved $X_{CH_4}$ values to correct for the systematic bias. As one fixed value of 0.977 is applied to all the TCCON sites, the site-to-site bias is not taken into account. It is assumed that the remaining systematic uncertainty of the TCCON $X_{CH_4}$ products is within 0.2%. The random uncertainty of the $X_{CH_4}$ retrieval is about 0.5% (Wunch et al., 2015). TCCON $X_{CH_4}$ observations have relatively larger footprints compared to surface in situ measurements, and thus could provide flux information on a regional scale (Wunch et al., 2016). TCCON $X_{CH_4}$ measurements are widely used to validate the satellite observations, e.g. the Scanning Imaging Absorption Spectrometer for Atmospheric Chartography (SCIAMACHY) and the Thermal And Near infrared Sensor for carbon Observation Fourier-Transform Spectrometer (TANSO-FTS) (Houweling et al., 2014; Dils et al., 2014; Zhou et al., 2016). In addition, the TCCON $X_{CH_4}$ observations are also used to evaluate the atmosphere chemistry transport model simulations (Saito et al., 2012; Fraser et al., 2013; Agusti-Panareda et al., 2017).

The concentration of atmospheric $CH_4$ remained almost constant from about 1995 to 2006. However, after 2007, the $CH_4$ concentration started to increasing with an annual growth rate about 0.7 ppb/year (Rigby et al., 2008). The $CH_4$ in the atmosphere is released from gas and oil, coal, landfill, ruminant animal, rice agriculture, biomass burning, wetland and lake. The $CH_4$ in the troposphere is removed mainly by the oxidation with hydroxyl radicals (OH), partly by the getting absorbed in the soil and party by reacting with chlorine radicals in the marine boundary layer. The $CH_4$ in the stratosphere is removed by the oxidation with OH, chlorine atoms and excited oxygen atoms (IPCC, 2013). The mole fraction of $CH_4$ decreases rapidly with altitude above the tropopause due to a higher photolysis rate in the stratosphere (Ehhalt and Heidt, 1973). The separation of the tropospheric and stratospheric $CH_4$ partial columns helps to better understand the atmospheric $CH_4$ variability and to comprehensively evaluate model simulations (Ostler et al., 2016; Saad et al., 2016; Wang et al., 2017). The seasonal variation of $CH_4$ in the troposphere is dominated by its source and sink as well as the horizontal transport, while seasonal variation of $CH_4$ in the stratosphere is strongly affected by the Brewer-Dobson circulation, the vertical transport, the intertropical convergence zone movement and stratospheric chemical reactions. A proxy method to derive the tropospheric and stratospheric $X_{CH_4}$ from the TCCON retrievals based on the known relationship between $CH_4$ and hydrogen fluoride (HF) or nitrous oxide ($N_2O$) in the stratosphere has already been demonstrated by Wang et al. (2014) and Saad et al. (2014). The $N_2O$ and HF total columns are also available in the TCCON standard products. An alternative $CH_4$ profile retrieval method has been provided by Tukiainen et al. (2016), using dimension reduction and the Markov chain Monte Carlo statistical estimation.

In this study, we employ the full-physics SFIT4 code to retrieve vertical profile of $CH_4$ from TCCON spectra (named SFIT4TCCON retrievals) at six sites (Ny-Ålesund, Sodankylä, Bialystok, Bremen, Orléans and St Denis) for measurements



performed during the time period of 2016-2017. The SFIT4 code is based on the optimal estimation method (Rodgers, 2000), which is an updated version of SFIT2 (Pougatchev et al., 1995) and commonly used in the Network for the Detection of Atmospheric Composition Change - the Infrared Working Group (NDACC-IRWG) (De Mazière et al., 2018). The TCCON sites and the SFIT4TCCON retrieval strategy are introduced in the next section. The motivation behind this study is to retrieve

vertical information of CH$_4$ from TCCON spectra. In section 3 the difference in X$_{CH_4}$ retrieved using the SFIT4TCCON to the X$_{CH_4}$ retrieved from the standard TCCON retrievals is investigated. The tropospheric and the stratospheric X$_{CH_4}$ retrieved using the SFIT4TCCON retrievals are compared with other available datasets, such as ground-based in situ measurements, Atmospheric Chemistry Experiment - Fourier Transform Spectrometer (ACE-FTS) satellite observations, and TCCON proxy data. Furthermore, the comparison results from the SFIT4TCCON retrievals relative to the AirCore profiles at Sondakylä

TCCON site (Kivi and Heikkinen, 2016) are also discussed in this section. Finally, conclusions are drawn in Section 4.

## 2 Data and method

### 2.1 TCCON sites

The locations of the TCCON sites used in this study are listed in Table 1. All sites use a Bruker IFS 125HR instrument to record NIR spectra in the range of 5000-10000 cm$^{-1}$ with a spectral resolution of 0.02 cm$^{-1}$. The TCCON spectra from all sites in

the time period of 2016-2017 were transferred to BIRA-IASB. A python code is developed to convert the TCCON spectra from the OPUS format to the SFIT4 readable format. A DC correction is applied to remove the noise of the interferogram caused by the solar intensity variation due to the presence of clouds during a measurement (Keppel-Aleks et al., 2007).

**Table 1.** The coordinates and the altitudes (m a.s.l.) of the TCCON FTIR sites used in this study.

| Site | Latitude | Longitude | Altitude (m a.s.l.) | Reference |
|---|---|---|---|---|
| Ny-Ålesund | 78.9°N | 11.9°E | 20 | Notholt et al. (2014b) |
| Sodankylä | 67.4°N | 26.6°E | 188 | Kivi et al. (2014) |
| Bialystok | 53.2°N | 23.0°E | 180 | Deutscher et al. (2014) |
| Bremen | 53.1°N | 8.8°E | 30 | Notholt et al. (2014a) |
| Orléans | 48.0°N | 2.1°E | 130 | Warneke et al. (2014) |
| St Denis (Reunion Island) | 21.0°S | 55.4°E | 87 | De Mazière et al. (2014) |

### 2.2 SFIT4TCCON retrieval strategy

The SFIT4TCCON retrieval strategy is investigated based on the TCCON spectra at St Denis using the SFIT4_v9.4.4 retrieval

code. After that, the optimized retrieval strategy is applied for other sites. The key parameters used in the SFIT4TCCON retrieval are listed in Table 2. The ATM spectroscopy used in the GGG2014 code (Toon and Wunch, 2014) has also been used in the forward model of the SFIT4TCCON. A linear polynomial fitting is applied to a time-domain ideal instrument line shape



(ILS) and the parameters are retrieved simultaneously. A detailed description of the retrieval settings, the averaging kernel and the retrieval uncertainty are presented in section 2.2.1 - 2.2.5.

**Table 2.** Lists of the most important parameters in the SFIT4TCCON $CH_4$ retrieval strategy.

| | |
|---|---|
| Retrieval window (cm$^{-1}$) | 5996.45-6007.55 |
| Interfering species | $CO_2$, $H_2O$ |
| Spectroscopy | ATM |
| Regularization | Tikhonov $L_1$ with $\alpha = 1000$ |
| A priori profile | WACCM v6 (fixed) |
| SNR | 250 |
| ILS | a linear polynomial fitting |

### 2.2.1  Retrieval window

Three windows which are listed in Table 3 are used to retrieve $CH_4$ total column values using the GGG2014 code. All these retrieval windows were tested with SFIT4TCCON retrieval. The typical transmittance and residual for the three windows are shown in Figure 1. The root-mean-square (RMS) of the residual in band 1 is largest due to a bad fitting of several strong $H_2O$ absorption lines. The RMS of the residual in band 2 is the lowest and is slightly better than that in band 3.

As an example, the retrieved $CH_4$ total columns using these three bands along with the dry air pressure (subtracting the water vapor pressure from the surface pressure) on 30 July 2016 are shown in Figure 2. A clearly artificial symmetric variation for band 1 is seen, which is probably due to a bad fitting of the spectra. As St Denis is a remote site and there is no strong $CH_4$ emission nearby, it is assumed that the diurnal variation of $X_{CH_4}$ above St Denis is relatively small. Therefore, the diurnal variation of $CH_4$ total columns is dominated by the variation of dry air columns ($TC_{CH_4} = X_{CH_4} \times TC_{air}^{dry}$). A high dry air pressure is expected to result in a high total column of dry air and vice versa. The correlation coefficients (R) between the dry air pressure and the retrieved $CH_4$ total columns from the band 1, 2 and 3 are 0.04, 0.82 and 0.27 respectively. In order to get a robust result, 6 more days in 2016 (01 March, 29 May, 30 June, 28 July, 20 September and 01 November) are selected for the analysis. The selection is made based on the criteria that there should be at least 100 measurements for that day and no strong diurnal variation in $X_{CH_4}$ is observed. The R between retrieved $CH_4$ total columns and dry air partial pressure is computed for each chosen day. The mean R value for band 2 is 0.78, which is much larger than the R value for band 1 (0.05) or the R value for band 3 (0.35). Consequentially, band 2 is selected as the retrieval window for our SFIT4TCCON retrieval.

### 2.2.2  A priori profile

According to the optimal estimation method (OEM) (Rodgers, 2000), a priori profile is used to initialize the iteration during a retrieval process. In this study, SFIT4TCCON uses the mean of the monthly profiles between 1980 and 2020 from the Whole





**Table 3.** The retrieval windows used in the GGG2014 code.

|        | Window ($cm^{-1}$) | Width ($cm^{-1}$) | Interfering spices |
|--------|---------------------|---------------------|----------------------|
| band 1 | 5781.0-5897.0 | 116.0 | $CO_2$, $H_2O$, $N_2O$ |
| band 2 | 5996.45-6007.55 | 11.1 | $CO_2$, $H_2O$ |
| band 3 | 6007.0-6145.0 | 138.0 | $CO_2$, $H_2O$ |

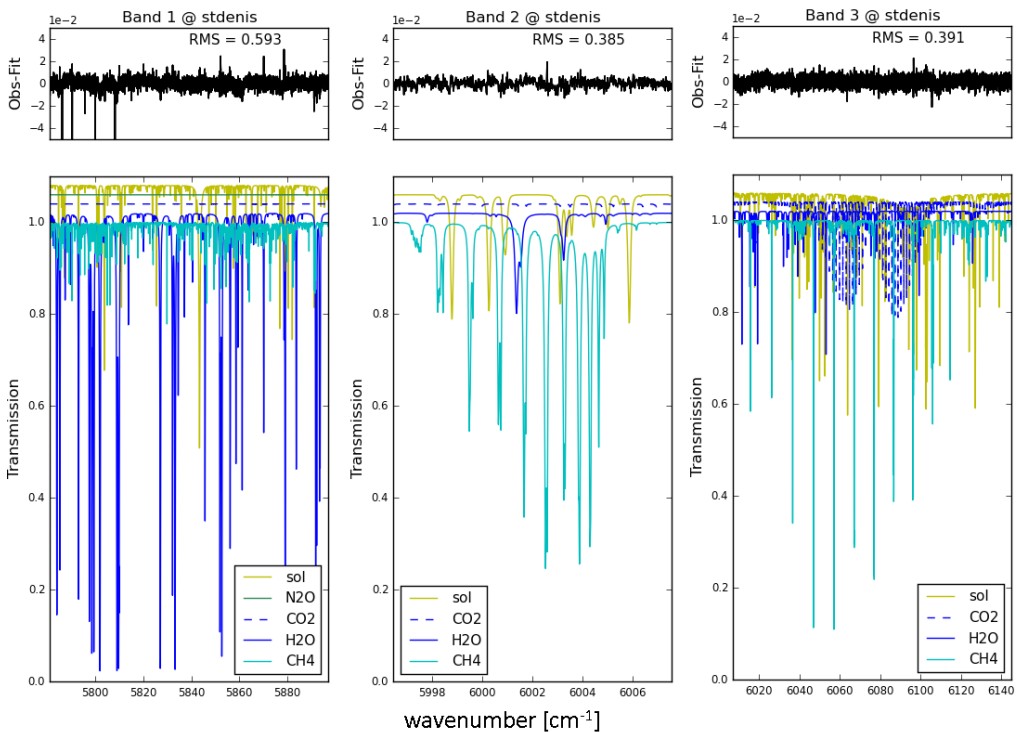

**Figure 1.** The transmittances of the species and solar lines in three bands from a typical spectrum at St Denis, together with the residual (observation - fitting). The transmittance of each component is shifted by 0.02 to better identify different species and the solar lines (sol).

Atmosphere Community Climate Model (WACCM) version 6 as the a priori profiles for $CH_4$ and $CO_2$, while $H_2O$ a priori profile is derived from the 6-hourly NCEP reanalysis data because of its high variability in the atmosphere.

### 2.2.3 Regularization

The retrieved $CH_4$ profile can be written as

$$x_{r,CH_4} = x_{a,CH_4} + \mathbf{A}(x_{t,CH_4} - x_{a,CH_4}) + \varepsilon, \tag{1}$$

$$\mathbf{A} = (\mathbf{K}^T \mathbf{S}_\varepsilon^{-1} \mathbf{K} + \mathbf{S_a}^{-1})^{-1} \mathbf{K}^T \mathbf{S}_\varepsilon^{-1} \mathbf{K}, \tag{2}$$





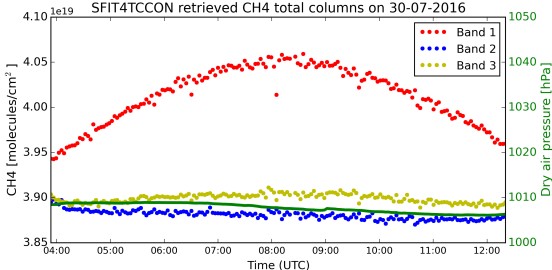

**Figure 2.** The SFIT4 retrieved $CH_4$ total columns using the 3 bands listed in Table 1 on 30 July 2016 (184 spectra), together with the computed dry air pressure for that day (green line).

where $x_{a,CH_4}$, $x_{t,CH_4}$ and $x_{r,CH_4}$ are the a priori, true and retrieved $CH_4$ mole fraction profiles, respectively. $\mathbf{A}$ is the averaging kernel, representing the sensitivity of the retrieved $CH_4$ profile to the true atmosphere status. The trace of $\mathbf{A}$ is the degree of freedom for signal (DOFS), indicating the number of individual vertical information derived from the retrieval. $\mathbf{K}$ is the Jacobian matrix. $\varepsilon$ is the retrieval uncertainty. $\mathbf{S_a}$ and $\mathbf{S}_\varepsilon$ are the a priori covariance matrix and the measurement covariance

5 matrix, respectively. $\mathbf{S}_\varepsilon$ is reply on the signal to noise ratio (SNR). $\mathbf{S_a}^{-1}$ and $\mathbf{S}_\varepsilon^{-1}$ are the two key parameters to constraint the retrieved $CH_4$, and to determine the retrieved $CH_4$ profile is mainly from the a priori information or from the measurement information. It is assumed that $\mathbf{S}_\varepsilon$ is a diagonal matrix, where the diagonal elements are the inverse square of the SNR. The SNR of spectra is set to 250 at all the TCCON sites. $\mathbf{S_a}^{-1}$ is created by the Tikhonov $\mathbf{L}_1$ method $\mathbf{S_a}^{-1} = \alpha \mathbf{L}_1^T \mathbf{T} \mathbf{L}_1 \in \mathbf{R}^{(n,n)}$ (Tikhonov, 1963). The matrix $\mathbf{T}$ considers the thickness of each layer. The regularization strength $\alpha$ is the key parameter to

10 control the strength of $\mathbf{S_a}^{-1}$.

The optimized $\alpha$ value is chosen by extracting maximum possible information from the measurement while eliminating the artificial oscillation for the retrieved $CH_4$ profiles. Several $\alpha$ values are tested using the spectra on 30 July 2016, and the RMS, DOFS and retrieved $CH_4$ total columns are listed in Table 4, along with the retrieved $CH_4$ vertical profiles in Figure 3. The retrieved $CH_4$ profile shows a strong oscillation in the troposphere for $\alpha = 100$. The vertical profiles are similar for $\alpha = 1000$

15 and $\alpha = 10000$, but it allows us to get a smaller RMS with $\alpha = 1000$. In summary, a regularization strength ($\alpha$) of 1000 with the DOFS of about 2.4 is the selected as the best choice for the SFIT4TCCON retrieval.

**Table 4.** The mean and standard deviation of RMS, DOFS and retrieved $CH_4$ total columns from SFIT4TCCON retrievals using different regularization strength $\alpha$ values.

| $\alpha$ | 100 | 1000 | 10000 |
|---|---|---|---|
| RMS (%) | 0.34±0.06 | 0.34±0.06 | 0.35±0.06 |
| DOFS | 3.25±0.11 | 2.42±0.08 | 1.77±0.04 |
| Total column $CH_4$ ($10^{18}$ molec/cm$^2$) | 38.62±0.05 | 38.64±0.04 | 38.65±0.04 |





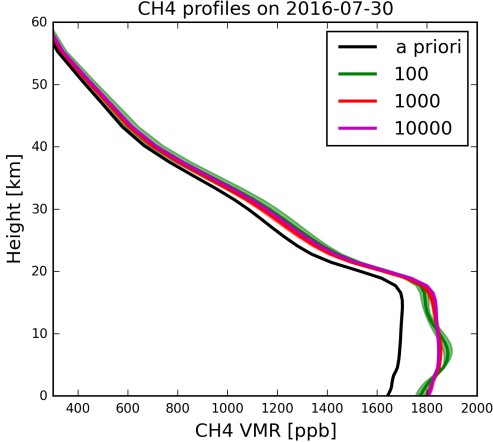

**Figure 3.** The SFIT4TCCON a priori and the retrieved $CH_4$ vertical profiles using different regularization strength $\alpha$ values.

### 2.2.4 Averaging kernel

Left panel in Figure 4 shows the typical averaging kernel (AVK) of SFIT4TCCON $CH_4$ retrieval with a solar zenith angle (SZA) of $63°$ at St Denis. The retrieved $CH_4$ profile is sensitive to the altitude range from the surface to the middle stratosphere (about 40 km). The AVK shows that the SFIT4TCCON retrieved profile contains independent information in the troposphere and in the stratosphere (DOFS close to 1.0 for these two layers). In addition, the column averaging kernels (right penal in Figure 4) indicate that the retrieved $CH_4$ total column has a good sensitivity in the whole atmosphere, with the value close to 1.0 at all altitudes. The column averaging kernels slightly vary with the SZAs, which is more constant than the AVK variability for the SZAs of the standard TCCON products (see Figure 4 in Wunch et al. (2011)).

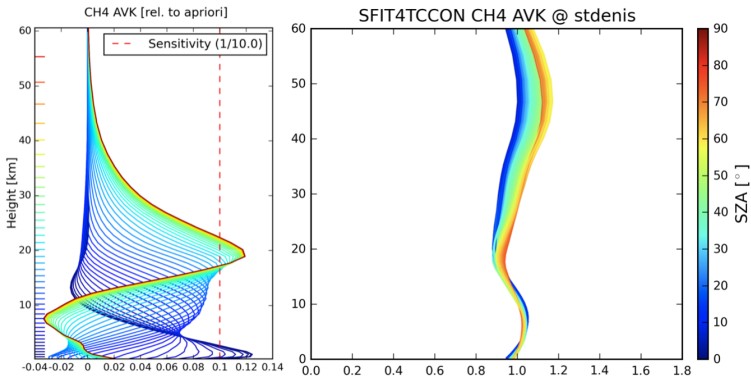

**Figure 4.** Left panel: a typical $CH_4$ averaging kernel matrix of the SFIT4TCCON retrieval with the SZA of $63°$ at St Denis, in unit of the mole fraction profile with respect to the a priori. Right panel: $CH_4$ column averaging kernels for different solar zenith angles.



### 2.2.5 Error budget

According to the OEM (Rodgers, 2000), the measurement uncertainty of SFIT4TCCON retrieval ($\varepsilon$ in Eq. 1) is estimated from three components: the smoothing error covariance matrix ($\mathbf{S}_s$), the forward model parameter error covariance matrix ($\mathbf{S}_f$) and the measurement error covariance matrix ($\mathbf{S}_m$).

$$\mathbf{S}_s = (\mathbf{A} - \mathbf{I})\mathbf{S}_e(\mathbf{A} - \mathbf{I})^T, \tag{3}$$

$$\mathbf{S}_f = \mathbf{G}_y\mathbf{K}_b\mathbf{S}_b\mathbf{K}_b^T\mathbf{G}_y^T, \tag{4}$$

$$\mathbf{S}_m = \mathbf{G}_y\mathbf{S}_\varepsilon\mathbf{G}_y^T, \tag{5}$$

where $\mathbf{G}_y$ is the contribution matrix, representing the sensitivity of the retrieval to the measurement. $\mathbf{S}_e$, $\mathbf{S}_b$ and $\mathbf{S}_\varepsilon$ are the covariance matrices of the retrieval state vector, the forward model parameter and the measurement, respectively. The retrieval state vector ($x$ in Eq. 1) not only includes the $CH_4$ vertical profile, but also includes the $H_2O$ and $CO_2$ columns, the slope of the background, the wavenumber shift and several ILS parameters. Each retrieved parameter has systematic and random uncertainties. The relative standard deviation of the $CH_4$ monthly means from the WACCM model in 1980-2020 is calculated as the random uncertainty of the $CH_4$ profile. For the systematic uncertainty, we have chosen a value of 5% (about 90 ppb in the troposphere), based on the difference between the a priori $CH_4$ mole fraction near the surface and the local in situ measurements (Zhou et al., 2018). As $CH_4$ is relatively stable in the atmosphere with a life time of $\sim$ 9 years, it is assumed that 5% systematic uncertainty is acceptable for all altitudes. The systematic and the random uncertainties for $H_2O$ and $CO_2$ are set to 5%. The systematic and random uncertainties of ILS parameters are set to 1%. The other retrieved parameters have do not contribute significantly to the $CH_4$ uncertainty. The smoothing error in Table 5 represents the uncertainty contribution from the $CH_4$ vertical profile, while the error from the retrieved parameters in the Table 5 include the contribution from the $H_2O$ and $CO_2$ columns, the slope of the background, the wavenumber shift and several ILS parameters. The spectroscopy, the temperature and the SZA are the most important parameters contributing to the forward model. According to the HITRAN2012 (Rothman et al., 2013), the uncertainty of $CH_4$ absorption in the selected retrieval window is about 2-5%. Therefore, the systematic uncertainty of the spectroscopy is set to 3%, and the random uncertainty on the spectroscopic data is assumed to be negligible. The systematic and random uncertainties are set to 1% for the temperature. The systematic uncertainty is set to 0.1% and the random uncertainty is set to 0.5% for the SZA. $\mathbf{S}_\varepsilon$ is assumed to be diagonal where the diagonal elements are the inverse square of the SNR. The propagated uncertainties of the total column and the partial columns (troposphere and stratosphere) are listed in Table 5. The mean tropopause height above St Denis is about 16.5 km. The systematic and random uncertainties of the SFIT4TCCON retrieved $CH_4$ total column are 3.2 and 0.5%, respectively. The dominating component of the systematic uncertainty is coming from the spectroscopy. The uncertainties of the partial column in the troposphere are closer to those of the total column, while the uncertainties of the partial column in the stratosphere are relatively large.





**Table 5.** The systematic and random uncertainties for the SFIT4TCCON retrieved CH$_4$ total column, partial columns in the troposphere (0–16.5 km) and in the stratosphere (16.5–50 km). The uncertainties are in the unit of percentage (%). The sign "–" is used for cases where the uncertainty is negligible with a value which is less than 0.1%.

| Error | Total column | | Troposphere (0-16.5 km) | | Stratosphere (16.5-50 km) | |
|---|---|---|---|---|---|---|
| | Systematic | Random | Systematic | Random | Systematic | Random |
| Smoothing | 0.2 | 0.2 | 0.2 | 0.2 | 1.2 | 1.6 |
| Measurement | – | 0.1 | – | 0.1 | – | 0.9 |
| Retrieved parameters | 0.2 | 0.2 | 0.1 | 0.1 | 2.5 | 2.5 |
| Temperature | 1.1 | 0.4 | 1.0 | 0.4 | 1.8 | 0.5 |
| Spectroscopy | 3.1 | – | 3.1 | – | 6.0 | – |
| SZA | 0.1 | 0.2 | 0.1 | 0.2 | 0.2 | 0.9 |
| Total | 3.2 | 0.5 | 3.1 | 0.5 | 6.8 | 3.3 |

## 3 Results and discussions

The retrieval strategy listed in Table 2 is applied for all six sites. In this section, the data quality of the SFIT4TCCON retrievals is evaluated with TCCON standard retrievals, ground-based in situ measurements, ACE-FTS satellite remote sensing observations, TCCON proxy X$_{CH_4}$ data and AirCore measurements.

## 3.1 TCCON standard retrievals

According to Section 2.2.5, the random uncertainty of SFIT4TCCON CH$_4$ total column is about 0.5%, which is close to that of TCCON retrieval (Wunch et al., 2015). The systematic uncertainty of SFIT4TCCON CH$_4$ total column is about 3.2% where a large contribution is from the spectroscopy. To better understand the systematic uncertainty of the SFIT4TCCON retrieved total column, the SFIT4TCCON X$_{CH_4}$ at six sites in 2016-2017 are calculated and compared with TCCON standard products. The systematic uncertainty of TCCON X$_{CH_4}$ products is within 0.2%.

GGG2014 uses the ratio between CH$_4$ and O$_2$ total columns to calculate the X$_{CH_4}$ (Yang et al., 2002), as the atmospheric O$_2$ mole fraction is relatively stable with the mole fraction of 0.2095

$$X_{CH_4} = 0.2095\, TC_{CH_4}/TC_{O_2}. \tag{6}$$

The X$_{CH_4}$ from the SFIT4TCCON retrieval is calculated as

$$X_{CH_4} = \frac{TC_{CH_4}}{TC_{air}^{dry}} = \frac{TC_{CH_4}}{P_s/(gm_{air}^{dry}) - TC_{H_2O}(m_{H_2O}/m_{air}^{dry})}, \tag{7}$$

where $TC_{air}^{dry}$ and $TC_{H_2O}$ are total columns of dry air and H$_2$O; $P_s$ is the surface pressure; g is the total column-averaged gravitational acceleration; $m_{H_2O}$ and $m_{air}^{dry}$ are molecular masses of H$_2$O and dry air, respectively (Deutscher et al., 2010). The





uncertainty of $P_s$ is better than 0.1 hPa and the uncertainty of $H_2O$ column in the troposphere is about 5-10%, as a result, the uncertainty of the dry air column is about 0.1%. The uncertainty of $X_{CH_4}$ is the combination of the uncertainties of the total column of $CH_4$ and the dry air column, while the uncertainty of the dry air column is negligible compared to the uncertainty of SFIT4TCCON $CH_4$ retrieval (see Table 5). The SFIT4TCCON tropospheric and stratospheric $X_{CH_4}$ can be calculated

following Eq. 7 using the partial columns of $CH_4$ and dry air in the troposphere and in the stratosphere, respectively. The tropopause height is calculated individually for each SFIT4TCCON retrieval using the temperature and pressure profiles from the NCEP 6-hourly reanalysis data. The tropopause height varies from site-to-site according to its latitude. The tropopause height is about 8–11 km at Ny-Ålesund and Sodankylä, 10–12 km at Bialystok, Bremen and Orléans, and 16–17 km at St Denis.

Figure 5 shows the time series of the hourly means of $X_{CH_4}$ from SFIT4TCCON and TCCON retrievals and their differences for measurements performed in 2016-2017. The mean and standard deviation of the $X_{CH_4}$ difference between SFIT4TCCON and TCCON (SFIT4TCCON - TCCON) at the six sites are in the range between -2.3 ppb (-0.14%) and 2.5 ppb (0.15%) and between 4.7 ppb (0.3%) and 9.7 ppb (0.5%). The standard deviations of the differences at all sites are within 0.5%, which is consistent with the combined random uncertainties from SFIT4TCCON and TCCON retrievals. The systematic bias

between the SFIT4TCCON and TCCON retrieved $X_{CH_4}$ is much lower than 3.2%, indicating that the systematic uncertainty of SFIT4TCCON $CH_4$ total column from the spectroscopy (see Table 3) is overestimated. Since the systematic uncertainty of TCCON $X_{CH_4}$ retrieval is better than 0.2%, it is inferred that the systematic uncertainty of SFIT4TCCON $X_{CH_4}$ retrieval is within 0.35%. Figure 6 shows the scatter plots of the $X_{CH_4}$ retrievals from SFIT4TCCON and TCCON at the six sites. The linear regression line (dashed red line) is very close to the one-to-one lines for all panels. The correlation coefficient is in the

range between 0.74 and 0.94. No obvious seasonal variation is seen from Figure 5 and 6.

## 3.2 In situ measurements

This section presents the comparison results between the ground-based in situ measurements at the individual sites and the tropospheric $X_{CH_4}$ retrieved using SFIT4TCCON. Ground-based in situ measurements are more sensitive to the local sources and sinks as compared to the FTIR measurements. The Traînou tower at Orléans site operates in situ measurements at four

heights (180, 100, 50 and 5 m). The measurements at 180 m is used here as it is less affected by the boundary layer (Schmidt et al., 2014). In situ measurements for the St Denis site is taken from the measurements performed at Maïdo (2155 m) located at about 20 km away from St Denis (Zhou et al., 2018). Both ground-based in situ measurements at Orléans and Maïdo are well calibrated frequently at the Laboratoire des Sciences du Climat et de l'Environnement (LSCE). The in situ measurements from other sites are not used in order to reduce the influence from the boundary layer.

Figure 7 shows the time series of the ground-based in situ measurements and the SFIT4TCCON tropospheric $X_{CH_4}$, together with their differences, at Orléans and St Denis. There are several high spikes seen in the in situ measurements, especially at Orléans, which are because of the influence from the boundary layer. The in situ measurements at Orléans are found to be about 36 ppb larger than the SFIT4TCCON tropospheric $X_{CH_4}$. Schmidt et al. (2014) showed that the $CH_4$ mole fractions at the 4 layers of the Orléans tower measurements are decreasing with increasing altitude. There is a strong $CH_4$ anthropogenic



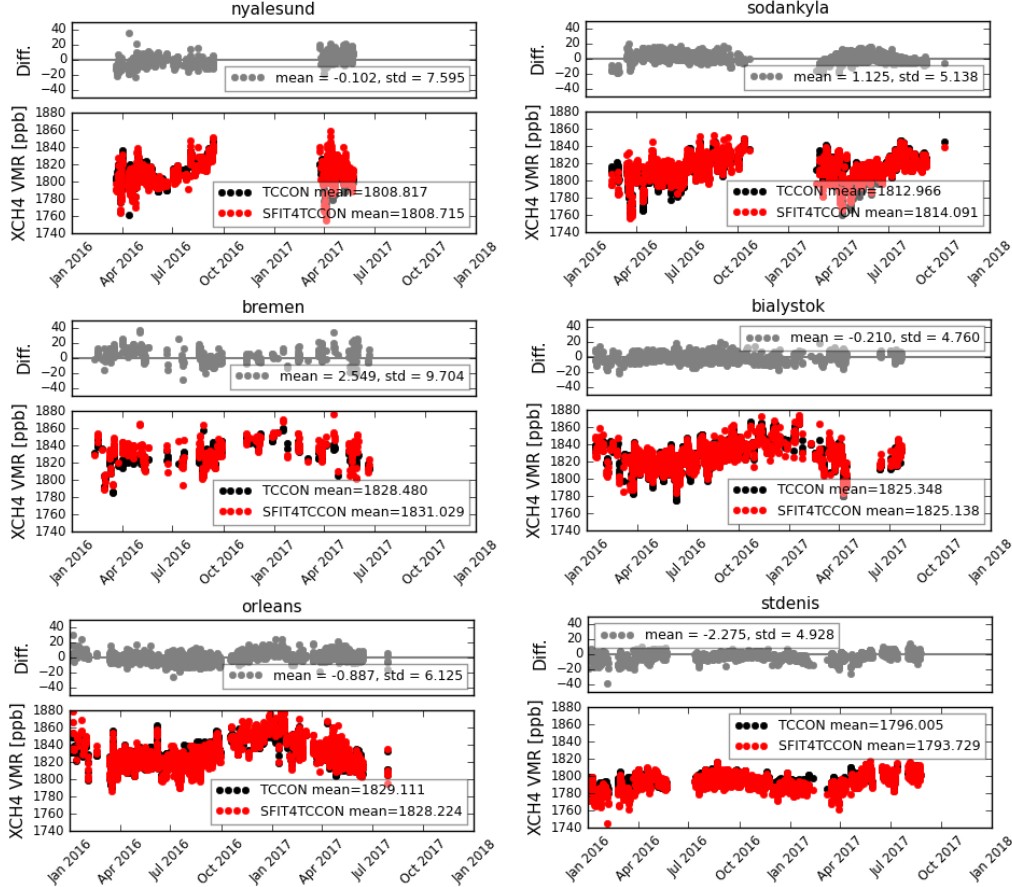

**Figure 5.** The time series of hourly means of $X_{CH_4}$ from the SFIT4TCCON and the TCCON retrievals at six TCCON sites during 2016 – 2017, together with their differences. For each site, the lower panel shows the time series of SFIT4TCCON and TCCON measurements, and the upper panel shows the absolute difference between them (SFIT4TCCON - TCCON; in ppb units).

emission around Orléans (European Commission, 2013), which remains mainly at the surface. This might explain the bias between the SFIT4TCCON tropospheric $X_{CH_4}$ and the in situ tower measurements at Orléans. The in situ measurements at St Denis are found to be about 24 ppb lower than the SFIT4TCCON tropospheric $X_{CH_4}$. Zhou et al. (2018) pointed out that the air near the surface above St Denis (0-2 km) is mainly coming from the Indian Ocean and partly from Southern African

5 region, whereas the air mass in the middle and upper troposphere (4-12 km) is mainly coming from Africa and South America. As $CH_4$ emission on the land is much larger than that from the ocean, it is reasonable that SFIT4TCCON tropospheric $X_{CH_4}$ is systematically larger than the $CH_4$ mole faction at the surface.

The phases and amplitudes of the seasonal cycles from the SFIT4TCCON tropospheric $X_{CH_4}$ and the ground-based in situ $CH_4$ measurements are found to be in good agreement. $CH_4$ mole fraction is high in January – March and low in July –

10 September at Orléans (located in Northern Hemisphere), and high in July – September and low in January – March at St Denis




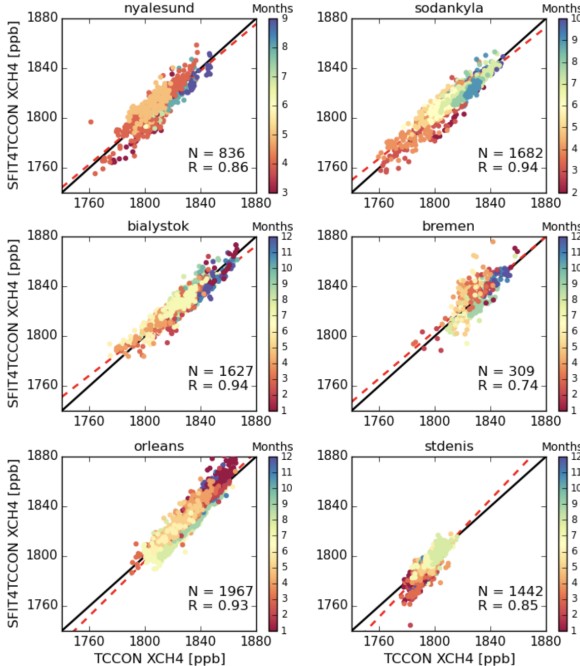

**Figure 6.** The scatter plots between SFIT4TCCON and TCCON $X_{CH_4}$ hourly retrievals at the six TCCON sites. The dots are colored with the measurement months. In each panel, the black line is the one-to-one line and the dashed red line is the linear fitting. N is the measurement number, and R is the correlation coefficient.

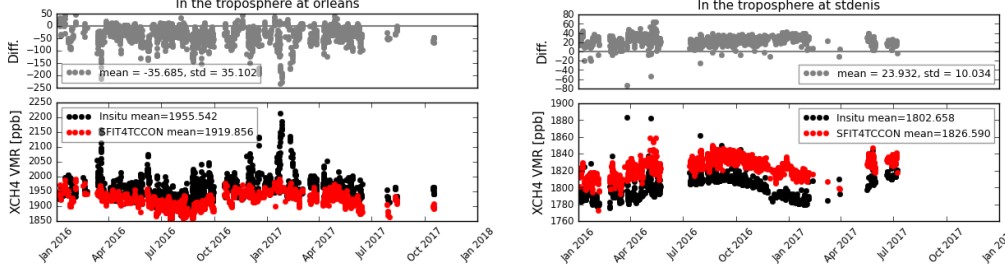

**Figure 7.** The time series of the hourly means from the SFIT4TCCON tropospheric $X_{CH_4}$ and the ground-based in situ $CH_4$ measurements at Orléans (left panels) and at St Denis (right panels), together with the absolute difference (unit: ppb) between them (lower and top, respectively). At Orléans, the in situ measurements are recorded at 180 m on a tower at the same place. The in situ measurements at St Denis are recorded at 2155 m on the Maïdo mountain, which is about 20 km away from St Denis.

(located in Southern Hemisphere). The $CH_4$ seasonal variations in the troposphere are driven by the OH variation, which is the major sink of $CH_4$ in the atmosphere.





### 3.3 ACE-FTS satellite observations

The comparison results between the SFIT4TCCON stratospheric $X_{CH_4}$ and the ACE-FTS satellite observations are discussed in this section. The vertical range from the tropopause height up to 50 km is treated as the stratosphere in this study. ACE-FTS satellite monitors the atmospheric $CH_4$ concentration mainly in the stratosphere since 2004 in the solar occultation mode

(Bernath et al., 2005). The latest level 2 version 3.6 data with data quality flag equal to 0 (without any known issue) are selected from ACE/SCIAST dataset (Sheese et al., 2015). The ACE-FTS $CH_4$ profile is retrieved on target altitudes with a vertical resolution of 3-4 km, and then it is interpolated onto a 1 km grid. To our knowledge, there is no validation report for the version 3.6 $CH_4$ data yet, but the older version v2.2 data of the ACE-FTS $CH_4$ data have been compared to space-based satellite, balloon-borne and ground-based FTIR data (De Maziere et al., 2008). The accuracy of the version 2.2 data is within

10% in the upper troposphere - lower stratosphere, and within 25% in the middle and higher stratosphere up to the lower mesosphere.

Figure 8 shows the SFIT4TCCON and ACE-FTS co-located daily means of the stratospheric $X_{CH_4}$ at Bialystok, Orléans and St Denis. The ACE-FTS measurements are selected within $\pm 3 \times 30°$ (latitude by longitude) around each FTIR site. Limited co-locations are found for Ny-Ålesund, Sodankylä and Bremen sites and so the results are not shown here. Figure 8 shows that

the seasonal cycles (both phase and amplitude) of the stratospheric $X_{CH_4}$ from SFIT4TCCON and ACE-FTS are similar. The stratospheric $X_{CH_4}$ shows a minimum in February-April and a maximum in August-October for the Bialystok and Orléans sites located in the northern hemisphere. Whereas, the stratospheric $X_{CH_4}$ shows a minimum in August-October and a maximum in February-April for St Denis site located in the southern hemisphere. The mean and the standard deviation of the differences in stratospheric $X_{CH_4}$ between the SFIT4TCCON and ACE-FTS measurements at these three sits are in the range between 0.27

and 2.06% and between 1.92 and 3.21%, respectively, which are within their uncertainties.

### 3.4 TCCON proxy data

In this section, the tropospheric and stratospheric $X_{CH_4}$ from SFIT4TCCON retrievals are compared with the results derived from the $N_2O$ and HF proxy methods at the six TCCON sites. We refer to Wang et al. (2014) and Saad et al. (2014) for the details of computing the tropospheric and stratospheric $X_{CH_4}$ by the proxy retrieval method using $N_2O$ and HF. Figure

9 shows the time series of the tropospheric and stratospheric $X_{CH_4}$ from the TCCON proxy $N_2O$ and HF method. First, the tropospheric and stratospheric $X_{CH_4}$ from the $N_2O$ and HF proxy methods are close to each other. However, a slight seasonal and site dependent bias is observed. The tropospheric $X_{CH_4}$ from $N_2O$ and HF proxy method are very close to each other for St Denis. While for the other five sites, the tropospheric $X_{CH_4}$ from the $N_2O$ method is larger by about 15-20 ppb than the HF method. The bias in the tropospheric $X_{CH_4}$ between the $N_2O$ and HF methods are in a good agreement with the Figure 7 in

Wang et al. (2014). As the TCCON $N_2O$ retrievals are corrected to the WMO scale (Wunch et al., 2015) while HF retrievals has not been validated, the systematic bias is probably due to the uncertainty of the $X_{HF}$ product. Second, at St Denis (a moist site), the TCCON HF retrievals are strongly affected by $H_2O$ so that the TCCON proxy method tropospheric and the stratospheric $X_{CH_4}$ data using HF have many outliers.





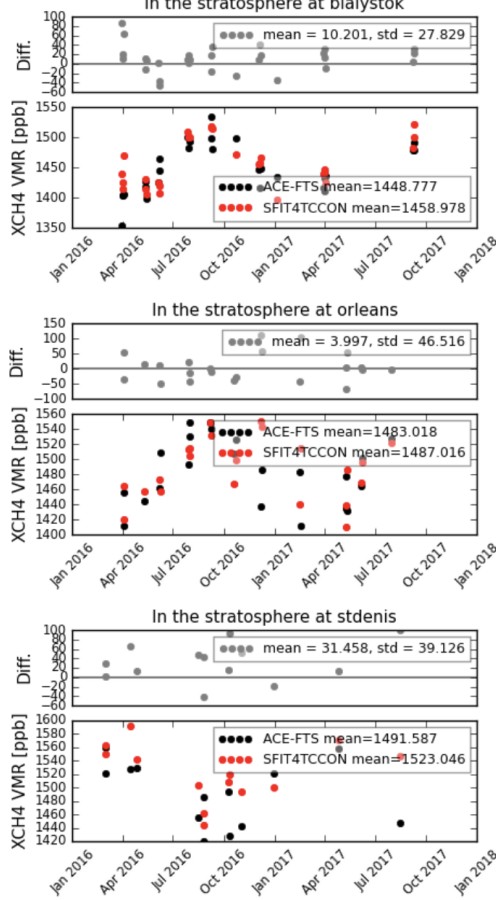

**Figure 8.** The time series of the daily mean of the stratospheric $X_{CH_4}$ from the SFIT4TCCON and the ACE-FTS co-located daily mean measurements, together with the absolute difference (unit: ppb) between them for Bialystok, Orléans and St Denis.

Figure 9 also shows the time series of the tropospheric and stratospheric $X_{CH_4}$ from the SFIT4TCCON retrievals. The SFIT4TCCON tropospheric $X_{CH_4}$ are close to the proxy data at St Denis, while the SFIT4TCCON tropospheric $X_{CH_4}$ are systematic larger than the results from the proxy method at the five sites located in the Northern Hemisphere. The vital difference between St Denis and other sites is that the tropopause height at St Denis is about 16.5 km, which is relatively higher

5   than the tropopause height of 9-12 km at other sites. It looks that the partial column of $CH_4$ from the SFIT4TCCON retrieval is larger/smaller than that from the TCCON proxy data in the vertical range from surface to about 10 km/above 10 km. The bias in the in the vertical range from surface to 10 km might be able to get compensation from the part of 10–16.5 km, due to the relatively high tropopause height at St Denis. The phases of the $X_{CH_4}$ seasonal variations from the SFIT4TCCON retrievals are almost the same as those from the proxy method, while the amplitudes of the variations from the SFIT4TCCON retrievals

10   are larger than those from the proxy method in the tropospheric component. There are several possible explanations: 1) the





proxy method assumes that the vertical mole fraction profile of HF or N$_2$O are constant in the troposphere; 2) the CH$_4$ mole fraction in the upper troposphere is calculated as the tropospheric X$_{CH_4}$ for the proxy method; 3) the tropopause height in the proxy method has a chemical definition, which differs from the tropopause height calculated from the temperature and the altitude profiles.

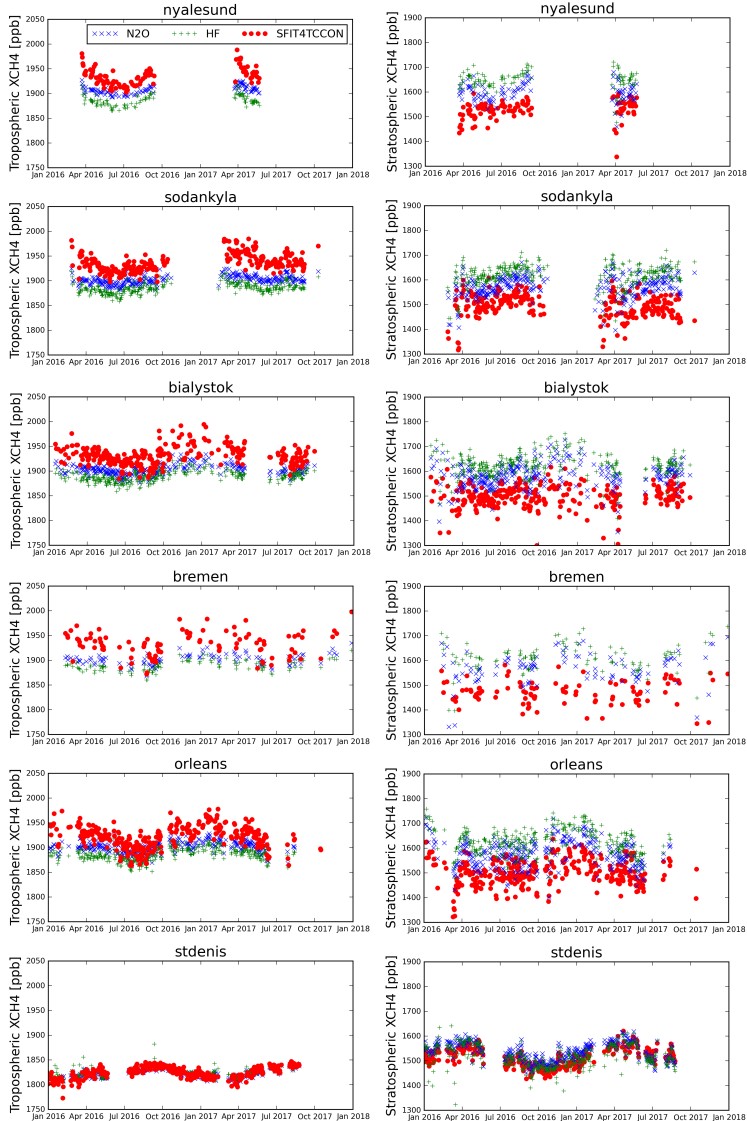

**Figure 9.** The time series of the tropospheric (left panels) and the stratospheric (right panels) X$_{CH_4}$ from the SFIT4TCCON and the proxy method (both N$_2$O and HF) at the six TCCON sites.





## 3.5 AirCore measurements at Sodankylä

The AirCore is an atmospheric sampling system which uses a long tube to sample the air from the surrounding atmosphere and preserve profiles of the trace gases of interest from the surface (few hundred meters) to the middle stratosphere (about 30 km) (Karion et al., 2010). Regular AirCore measurements of $CH_4$ have been carried out at Sodankylä since September 2013.

During 2016-2017, we select 7 AirCore profiles which are within 1 hour of SFIT4TCCON measurements.

As an example, the AirCore $CH_4$ profile on 5 September 2017, together with the co-located (within $\pm$ 1 hour) SFIT4TCCON a priori and retrieved profiles are shown in the left panel of Figure 10. The AirCore measurement for this launch only covers the vertical range from 0.6 km to 26 km, and needs to be extended to compare with the FTIR data. A scaled SFIT4TCCON a priori profile is applied to extend the AirCore $CH_4$ profile above 26 km. The local surface $CH_4$ mole fractions are observed

at 0.048 km by a Picarro G2401 instrument at Sodankylä (Kilkki et al., 2015). A linear interpolation is applied based on the AirCore measurement and the simultaneous in situ measurement to obtain the $CH_4$ mole fraction below 0.6 km. The "extended" AirCore profile is then smoothed with the closest SFIT4TCCON retrieval. The mean and the standard deviation of the relative differences between the co-located SFIT4TCCON retrievals and the smoothed AirCore profile ((SFIT4TCCON - AirCore) / AirCore $\times$ 100%) are shown in the right panel of Figure 10. The bias is about +1.5% in the lower and the middle

troposphere, between +1 and -4% in the upper troposphere and lower stratosphere region, and about -2.5% in the middle and upper stratosphere.

Figure 11 shows the scatter plots of $X_{CH_4}$ between the co-located SFIT4TCCON retrievals and the AirCore measurements for the whole atmosphere, for the tropospheric and for the stratospheric components. The errorbars are the random uncertainties of the SFIT4TCCON retrievals and the AirCore measurements. It is assumed that the random uncertainty of the AirCore profile

is about 0.1% between the surface and its maximum measurement altitude ($\sim$ 30 km), and it is about 2% above the maximum measurement altitude. The slope of the regression line (a = 1.001) in the whole atmosphere indicates that there is almost no systematic difference between the SFIT4TCCON and the AirCore $X_{CH_4}$, which is consistent with the result in the comparison between SFIT4TCCON and TCCON $X_{CH_4}$ measurements (Figures 5 and 6). The SFIT4TCCON tropospheric $X_{CH_4}$ is about 1.2% larger than the AirCore measurements and the SFIT4TCCON stratospheric $X_{CH_4}$ is about 4.0% less than the AirCore

measurements. These differences between the SFIT4TCCON retrievals and AirCore measurements are within the systematic uncertainties of the SFIT4TCCON partial columns in the troposphere and stratosphere, and it is inferred that the systematic uncertainty of the SFIT4TCCON partial column is mainly coming from the uncertainty of the spectroscopy (see Table 5). Further investigation is required to see if the systematic bias can be observed at other sites.

## 4 Conclusions

The retrieval of $CH_4$ vertical information from TCCON FTIR spectra has been carried out at six sites during 2016-2017 using the SFIT4 code. The retrieval strategy of the SFIT4TCCON has been discussed, including the spectroscopy, retrieval window, a priori profile, SNR and regularization. The AVK shows that the SFIT4TCCON retrieved profile is sensitive to the altitude range from the surface to the middle stratosphere (about 40 km), and the column averaging kernel has a good sensitivity in the





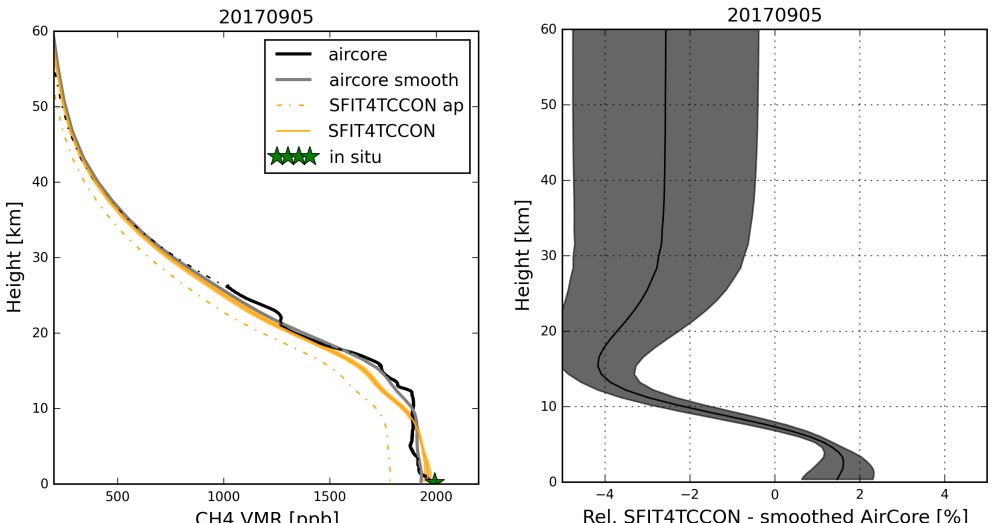

**Figure 10.** Left panel: the $CH_4$ profile from the AirCore measurement (solid black line) on 5 September 2017, together with the SFIT4TCCON a priori (dash-dotted orange line) and retrieved (solid orange line) profiles. The AirCore measurement is extrapolated with the surface in situ measurements (green star) and the scaled SFIT4TCCON a priori profile (dotted black line). The grey line is the smoothed AirCore profile. Right panel: the mean (solid black line) and the stand deviation (shadow) of relative difference between the co-located SFIT4TCCON retrieved profiles and the smoothed AirCore measurements ((SFIT4TCCON - AirCore) / AirCore × 100%).

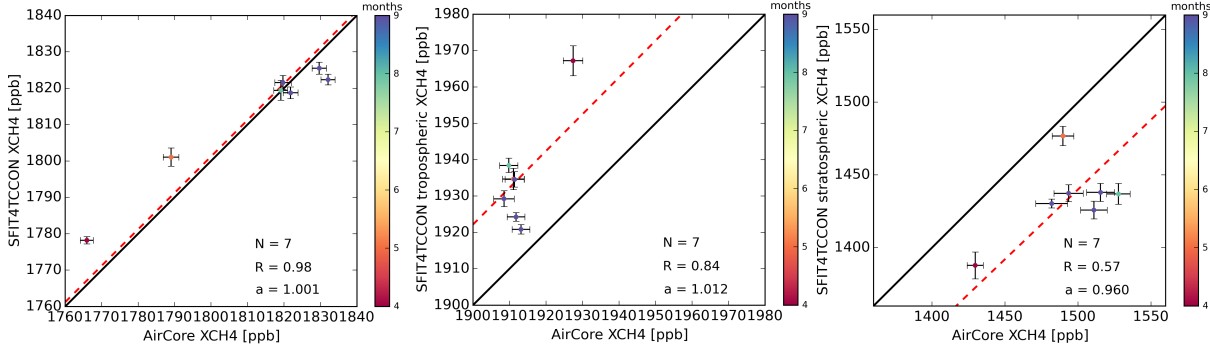

**Figure 11.** The scatter plots of $X_{CH_4}$ between the SFIT4TCCON and the AirCore measurements for the whole atmosphere (left panel), for the tropospheric (middle panel) and for the stratospheric (right panel) components. In each panel, the black line is the one-to-one line and the dashed red line is the regression line with the intercept to zero ($y = a \cdot x$). N is the co-located measurement number, R is the correlation coefficient, and a is the slope.

whole atmosphere. The DOFS of the SFIT4TCCON is about 2.4, with two distinct pieces of information in the troposphere




and the stratosphere. The systematic and random uncertainties of the SFIT4TCCON retrieved total column are about 3.2 and 0.5%.

The SFIT4TCCON retrieved $CH_4$ total columns and partial columns (troposphere and stratosphere) have been evaluated based on the standard TCCON retrievals, ground-based in situ measurements, ACE-FTS satellite observations, TCCON proxy $X_{CH_4}$ data, and AirCore measurements at Sodankylä. It is found that the SFIT4TCCON retrieved $X_{CH_4}$ data are very close to the standard TCCON retrievals with the mean bias between -0.14 and 0.15% and the standard deviation of bias between 0.3 and 0.5% at the six TCCON sites. Additionally, there is no obvious seasonal variation in the difference between the SFIT4TCCON and TCCON $X_{CH_4}$ data. The SFIT4TCCON tropospheric and stratospheric $X_{CH_4}$ can observe the $CH_4$ seasonal variation very well, which has been confirmed by the ground-based in situ measurements and ACE-FTS observations, respectively. The tropospheric and the stratospheric $X_{CH_4}$ from SFIT4TCCON retrievals have also been compared with the results from the TCCON proxy method. The phases of the seasonal cycles from SFIT4TCCON retrievals and TCCON proxy data are consistent, though the amplitudes of the variations from the SFIT4TCCON retrievals are relatively larger than those from the proxy method, especially in the troposphere. As there are several limitations in the assumption of TCCON proxy method and the seasonal cycle from the tropospheric $X_{CH_4}$ from the SFIT4TCCON retrieval is very close to that from the tower measurements at Orléans, it is inferred that the seasonal cycle from the tropospheric $X_{CH_4}$ is more reliable. Further investigation is needed to understand the accuracy of the seasonal cycle from the SFIT4TCCON tropospheric $X_{CH_4}$ when more aircraft or AirCore measurements become available. By comparison against AirCore measurements at Sodankylä, it is found that there is almost no systematic bias between the SFIT4TCCON and AirCore $X_{CH_4}$, which is consistent with the comparison between the SFIT4TCCON and the TCCON standard retrievals. An overestimation of 1.2% in the SFIT4TCCON tropospheric $X_{CH_4}$ and an underestimation of 4.0% in the SFIT4TCCON stratospheric $X_{CH_4}$ is seen by comparing with AirCore measurements. These values are within the systematic uncertainties of SFIT4TCCON retrieved partial columns in the troposphere and stratosphere, respectively.

*Data availability.* The TCCON data are publicly available through the TCCON wiki (https://tccondata.org/). The ACE-FTS data used in this study are available from http://ace.uwaterloo.ca/data/ (registration required). The SFIT4TCCON retrievals, TCCON proxy data, ground-based in situ measurements and AirCore measurements are available by contacting the author.

*Competing interests.* The authors declare that they have no conflict of interest.

*Acknowledgements.* This study is supported by the EU H2020 RINGO project. The TCCON site at Réunion Island is operated by the Royal Belgian Institute for Space Aeronomy with financial support in 2014, 2015, and 2016, 2017 under the EU project ICOS-Inwire and the ministerial decree for ICOS (FR/35/IC2) and local activities supported by LACy/UMR8105 - Université de La Réunion. We want to thank Francis Scolas (BIRA-IASB) for maintaining the TCCON measurements at St Denis, Juha Hatakka and Tuomas Laurila (FMI) for operating



the in situ and AirCore measurements at Sondakylä. We would also like to thank TCCON network for making the data publicly available. The AirCore data from the Sodankylä site in 2017 is generated from the FRM4GHG project which received financial support from the European Space Agency under the grant agreement number ESA-IPL-POE-LG-cl-LE-2015-1129.

*Author contributions.*   MZ wrote the manuscript. MZ, BL and MDM investigated the SFIT4 retrieval strategy. MKS, CH, CP, TW, JMM, RK and PH provided and interpreted the TCCON data. NK and MR provided and analysed the in situ measurements. HC and RK provided the AirCore measurements. All authors read and commented the manuscript.




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
