# Peer review of "Retrieval of atmospheric CH4 vertical information from ground-based FTIR near-infrared spectra"

_Atmospheric Measurement Techniques, 2019_

## Referee Comment (RC1) · Anonymous Referee #2 · 1 Jul 2019

The study by Zhou et al. employs the full-physics retrieval code SFIT4, which is used by the Network for the Detection of Atmospheric Composition Change (NDACC) in order to retrieve vertical profile information on atmospheric methane from solar absorption spectra measured in the near infrared (NIR) by spectrometers within the Total Carbon Column Observing Network (TCCON). Comparisons of retrieval codes lead to improvements in the codes and therefore, this study is a contribution to remote sensing measurements of atmospheric CH4. I recommend its publication in AMT after the questions, issues and comments outlined below have been addressed.

Major Comments:

[Figure]

The authors state that the ILS parameters are retrieved simultaneously by the code. How does the retrieved instrument line shape look like and how constant is it for all the sites involved? The Bruker 125HR spectrometers exhibit excellent ILS stability, so the retrieved values should reflect this. Therefore, it would be beneficial if the authors could show a time series of the ILS and the parameters.

The profile retrieval relies on the Alpha values, as discussed in Sec 2.2.3, but could the authors please explain the physical significance of the Alpha value? Also, it seems to me, as shown in Fig. 3, that the retrieved profile just approaches a scaled value of the a-priori profile at Alpha values of 1,000 and 10,000. Optional addition to Fig 3: Could the authors add a plot of retrieved VMR profile divided by the a-priori VMR profile with altitude as y-axis or something similar? This is to show how much the a-priori is scaled and the altitude dependence of this value.

In TCCON, the Xair value and its time series are indicative of instrument stability, I think a comparison of the SFIT retrieved Xair and the TCCON Xair for the sites is warranted for this study.

In situ measurements: In its current state, I do not see the full usefulness of the comparison between the in situ ground-based measurements and the tropospheric product of SFIT4TCCON (Sec 3.2). Both measurements have completely different sensitivities, as the authors mentioned, and I think comparing the time-series alone does not sufficiently provide information to say that "The SFIT4TCCON tropospheric and stratospheric XCH4 can observe the CH4 seasonal variation very well, which has been confirmed by the ground-based in situ measurements. . ." in the conclusions. For example, the agreement between SFIT4TCCON tropospheric CH4 and in situ looks to be closer during the winter months and farther during the summer months both at Orleans and St. Denis. But it is difficult to see from the scattered, overlapping data points. I recommend that the authors derive a seasonal fit and/or trend line to both FTS and in situ data. Alternatively, a correlation/scatter plot for the FTS vs. in situ would be useful, like in Fig 6.

Finally, it wasn't clear to me until the last sentence of Sec 2.2.1 that the SFIT4TCCON retrieval does not actually use all the CH4 bands used by TCCON, so the term "SFIT4TCCON" is potentially misleading to the readers and data users. I recommend that this term be changed. Moreover, the authors state in the abstract that "the SFIT4 retrieval code is applied to retrieve CH4 mole fraction vertical profile using TCCON spectra". First, this is not entirely true because the retrieval only uses a subset of the NIR spectra used for retrieval of CH4 in the TCCON network. Second, the term "TCCON spectra" is a term that is not officially used by TCCON and this is also not a standard TCCON product, so I do not think that this term is appropriate to be used to describe the spectra used in this study.

Technical/Minor Comments:

Figure axes labels: The "4" in CH4 should be subscripted when possible.

P1, Line 7: two distinct species -> two distinct pieces

P1, Line 19: spectrometer -> spectrometers

P2, Line 16: the atmosphere chemistry -> atmospheric chemistry

P2, Line 23: started to increasing -> started to increase

P2, Line 24: remove "the" in: partly by the getting -> partly by getting

P3, Line 15: It seems that the spectra is converted to SFIT4 readable format then corrected for SIV, please arrange sentences if not the case.

Figure 2. How do the retrieved columns for each band look like for the standard TCCON product? Does the TCCON CH4 at band 1 also exhibit the same curve?

P6, Line 3: The DOFS definition is not entirely accurate; moreover it is not consistent with line 7 of the Abstract.

P6, Line 5: The sentence about S_(epsilon) needs to be checked, it seems wrong.

P6, Line 5: change "to constraint" to "to constrain"

P6, Line 5: "to determine whether the" or "to determine if the"

P7, Line 5: change "penal" to "panel"

P8, Line 17: remove "have" in "parameters have do not"

P8, Line 19: change "error" to "errors"

P10, Lines 25 and 26: use plural "measurements .... are "

P10, Line 28: Either quantify how well they are calibrated or change "well calibrated" to just "calibrated"

Fig. 5: The panels are too crowded and the legend boxes partially cover the data. I think this should be improved. The TCCON and SFIT4TCCON data overlap and it is impossible to determine the data points. This figure needs to be revised. The same goes for Fig. 7.

Fig. 7, Legend: change "Insitu" to "in situ". Title: fix "stdenis" and "orleans"

Fig 8: The legends overlap with the actual data points, making it hard to read the figure. Additionally, Fig. 8 could be improved by adding a correlation plot for each panel because the data are too sparse and does not cover a long time series. In fact, the correlation plots could be a better representation.

P13, Line 9: There have been other validation activities after De Maziere et al., 2008 and the results for CH4 have improved since then, e.g. https://doi.org/10.4401/ag-6339

P13, Line 19: "sits" to "sites"

Figure 9: I would like to know how the data points in Fig. 9 are treated. Daily means, hourly means? How is the filtering done and how are the errors in each retrieval taken into account? The answer to this could explain or support the statement "at St Denis (a moist site), the TCCON HF retrievals are strongly affected by $H_2O$ so that the

TCCON proxy method tropospheric and the stratospheric XCH4 data using HF have many outliers"

P13, Line 26-27: This "slight seasonal and site dependent bias" is not clear to me from the figure.

P14, Line 3: "systematic larger" -> "systematically larger"

P14, Line 5: The sentence starting from line 5 and ending in line 6 needs to be improved.

Fig. 10 Caption: it is very hard to see the "scaled SFIT4TCCON a priori profile (dotted black line)"

Fig. 10 and Sec 3.5: Why and how is the AirCore profile smoothed? It seems that there is a lot of structure and information in the AirCore profile that is lost from the smoothing. Also, the sentence "The extended" AirCore profile is then smoothed with the closest SFIT4TCCON retrieval" is not very clear.

Fig. 11: The authors need to provide an error estimate on the slopes of the lines.

P18, Lines 13-15: I think measurements from a single tower compared to a single TCCON site are not sufficient to arrive at this conclusion. Moreover, the authors should quantify what they mean by "very close".

P18, lines 17-20. These sentences seem contradicting. On one hand the authors state that "there is almost no systematic bias between the SFIT4TCCON and AirCore XCH4", but on the other hand the next sentence state "An overestimation of 1.2% in the SFIT4TCCON tropospheric XCH4 and an underestimation of 4.0% in the SFIT4TCCON stratospheric XCH4 is seen by comparing with AirCore measurements"

---

## Referee Comment (RC2) · Anonymous Referee #1 · 20 Jul 2019

The manuscript "Retrieval of atmospheric CH4 vertical information from TCCON FTIR spectra" by Zhou et al. uses FTIR spectra recorded at six different TCCON sites and applies an alternative data processing: instead of column-averaged dry air mole fractions of CH4, altitude profiles (or rather partial columns) of tropospheric and stratospheric CH are retrieved. The results are compared to several other data sets for validation:

- standard TCCON CH4 column data product

- stratospheric profiles from ACE-FTS

- a proxy used in TCCON to separate tropospheric and stratospheric partial columns

- tropospheric and stratospheric in-situ CH4 profiles at one site

[Figure]

In my opinion, the manuscript addresses an important issue: a profile data product from TCCON spectra hae been requested by several TCCON data users in the past. So, in principle, this study could be very useful to evaluate the possibilities and limitations of such a data product.

Unfortunately, the manuscript fails at this attempt. While the retrieval itself looks fine, the validation of the method is flawed. The most obvious way to validate the method would have been comparing the retrieved profiles to in-situ aircraft and balloon (aircore) profiles. Dozens of such profiles have been measured at various TCCON stations during the last 10-15 years and are available for the TCCON community. However, the authors decided to select six sites of which only one actually had in-situ profiles during the - also arbitrarily chosen - time period. Why did they not use TCCON sites that had aircraft profiles in the past and processed the spectra from those days? It would have been easy to apply the profile retrieval to other time periods. I understand that they did not want to process many more years of TCCON data. But I also do not see a need for using the same time period for all stations in the first place.

As a result, they struggle with the validation of their retrieved stratospheric profiles. They resort to using satellite CH4 profiles from ACE-FTS where the precision and accuracy is more than an order of magnitude worse than the TCCON standards for column retrievals. Instead, they could have done an extensive comparison with data from the Sodankylä site but for an extended time period. And there are more TCCON sites that have been covered by aircore measurements than just Sodankylä. One is even part of this study: Orleans!

In the troposphere, it is even more puzzling why the bulk of available in-situ aircraft profile data was completely ignored. Even during the same time period, several TCCON sites were overflown by the NASA ATom campaign. And several of the sites that have been used in this study had aircraft profiles taken during the IMECC campaign in 2009. Why did they not process spectra from that time period for these sites instead? There is no obvious reason why they limited themselves to 2016-17.

Given these issues, I cannot recommend publication without major revisions. The authors should try their profile retrieval on stations and time periods with corresponding aircraft or aircore in-situ CH4 profiles. If they do so, the whole section 3.3 (ACE-FTS comparison) could be dropped. The section 3.4 (TCCON proxy) would still be useful to evaluate if the proxy method or a profile retrieval provides better results.

General comments:

Why were these sites selected? Among those, Sodankylä is the only one where in-situ profiles were used to validate the retrieval. Others sites do have profiles but they were ignored. To my knowledge, two sites never had any kind of in-situ profiling but were included nonetheless.

Aircraft/aircore profiles at the selected sites:

- Sodankylä: many during several campaigns (but only a limited set used here)

- Bialystok, Bremen, Orléans: aircraft profiles taken 2009 but not used here. Data published in Geibel et al.: Calibration of column-averaged CH4 over European TCCON FTS sites with airborne in-situ measurements, Atmos. Chem. Phys., 12, 8763-8775, https://doi.org/10.5194/acp-12-8763-2012, 2012.

- Ny-Alesund: never

- St Denis: never

ATom visited several TCCON sites in 2016/17, the data is publicly available. Why was this data set completely ignored?

Why was the time frame 2016-17 selected? There is no special reason given and I cannot see one that requires to try the SFIT4TCCON on the same time period for all sites. It would make much more sense to use it on persiods with available in situ profiles for each site.

Why was St. Denis used as the prototype? It is probably the wettest site of all and not

necessarily characteristic for the others.

Specific comments:

The data product is called SFIT4TCCON but - unlike the name suggests - it is not an official TCCON data product. Was the use of the name "TCCON" approved by the TCCON PIs? I am a TCCON PI myself but I am not aware of this.

Section 2.2.1:

- p. 4, l. 11: Diurnal variations above St Denis are assumed to be small. What about the biomass burning season, would this still be true? At least there should not have been much biomass burning activity during the chosen time frame. It would have been easy to check other sites TCCON sites for these criteria. Also, I do not understand why the band with the highest correlation between CH4 and dry air pressure is best choice.

- p. 6, l. 8-10: the "$S_a^{-1}$ = ..." should be a numbered equation. The quantity "L1" is not described at all, "T" only very briefly. Provide a reference linking Tikhonov 1963 to the Sa matrix. In the cited reference, Tikhonov certainly did not refer to Rodger's original work which came years later (1976).

Section 2.2.4:

- How was the SFIT4 column AVK calculated?

Section 2.2.5:

- 5% uncertainty might be fine for St Denis sources but may underestimate the variability closer to CH4 sources. Check available aircraft profiles (e.g. from ATom, HIPPO etc.) as well as aircore to get a better grip on the expected variability. The model data might be too smooth.

Eq. 7: Why did you use pressure and not the O2 column for the calculation dry air column? There are good reasons why TCCON abandoned surface pressure as a proxy for airmass years ago.

Section 3.2

- It is doubtful that surface in-situ measurements from 20 km away are representative for St Denis. If you think otherwise, you should provide some reasonable arguments.

Section 3.3

- I understand the need to find something to compare the stratospheric profiles with. However, given the very limited precision and accuracy listed here for the 10-year old ACE-FTS data (25%), I wonder how useful this comparison can be. One might be better off comparing to zonal means with better statistics. That might also provide profiles for the stations where reasonable co-locations could not be found.

- If you decide to make the major revisions that I recommend, this whole section could be dropped.

Section 3.4

- Due to the H2O interference in the HF microwindow, I would trust the N2O method more for a tropical site like St. Denis.

- p. 14, l. 10 to p. 15, l. 4: it should not be too difficult to find out which of the raised possibilities 1) to 3) are true. The source code for the TCCON retrieval as well as the handling of the a priori profiles is available.

Section 3.5

- I believe that the Aircore measurements are the most trustworthy validation source for the tropospheric and stratospheric profiles. There are other TCCON sites which had simultaneous aircore profiles taken. Why were these not used? This would also improve the statistics in Fig. 11, which suffers from the fact that a very limited range of XCH4 was compared.

References:

- All the TCCON dataset citations are wrong: they still refer to Oak Ridge Naional Laboratory from which the TCCON data archive moved away already in October 2017! Strange how this can happen when several TCCON PIs are listed as coauthors.

- Acknowledgments: the authors should check if the acknolwedgments are in line with what is required by the TCCON Data Use Policy.

Minor comments:

- p. 3, l. 15: I would not call that "noise".

- p. 3, l. 21: Please explain the ATM acronym at least once.

- Fig. 2: poor color choice for dry air pressure as green is already used for band 3.

- p. 6, l. 5: should be "constrain" instead of "constraint"

- p. 7, l. 5: should be "panel" instead of "penal"

- p .8, l. 17: "The other retrieved parameters have do not contribute ..."

- p. 8, l. 28: "retrieved CH 4 total column are 3.2 and 0.5%" units missing

- Table 5: Why not just write "<0.1%" instead of "-"?

- p. 9, l. 2: "applied to" instead of "applied for"

- p. 11, l. 1: Are there really no authors and no better reference for the EDGAR database?

- Eq 6: TC is probably total column but should be explained

- p. 14, l. 3: "systematically" instead of "systematic"

- p. 14, l. 4: drop "relatively". It is also absolutely higher.

- p. 14, l. 5-6: the sentence started in line 5 should end with ", respectively."

- p. 18, l. 23: The URL points to the TCCON Data Archive, not to the TCCON wiki.

---

## Author Comment (AC2) · 13 Sep 2019

*Black: referee's comments* *red: authors' answers*
*First of all, we want to thank the two referees for the detailed analysis of our paper.*
*For the details, please look into the paper with keeping track of changes.*

Anonymous Referee #1

The manuscript "Retrieval of atmospheric CH4 vertical information from TCCON FTIR spectra" by Zhou et al. uses FTIR spectra recorded at six different TCCON sites and applies an alternative data processing: instead of column-averaged dry air mole fractions of CH4, altitude profiles (or rather partial columns) of tropospheric and stratospheric CH are retrieved. The results are compared to several other data sets for validation:
- standard TCCON CH4 column data product
- stratospheric profiles from ACE-FTS
- a proxy used in TCCON to separate tropospheric and stratospheric partial columns
- tropospheric and stratospheric in-situ CH4 profiles at one site
In my opinion, the manuscript addresses an important issue: a profile data product from TCCON spectra have been requested by several TCCON data users in the past. So, in principle, this study could be very useful to evaluate the possibilities and limitations of such a data product.

Unfortunately, the manuscript fails at this attempt. While the retrieval itself looks fine, the validation of the method is flawed. The most obvious way to validate the method would have been comparing the retrieved profiles to in-situ aircraft and balloon (aircore) profiles. Dozens of such profiles have been measured at various TCCON stations during the last 10-15 years and are available for the TCCON community. However, the authors decided to select six sites of which only one actually had in-situ profiles during the - also arbitrarily chosen - time period. Why did they not use TCCON sites that had aircraft profiles in the past and processed the spectra from those days? It would have been easy to apply the profile retrieval to other time periods. I understand that they did not want to process many more years of TCCON data. But I also do not see a need for using the same time period for all stations in the first place.

As a result, they struggle with the validation of their retrieved stratospheric profiles. They resort to using satellite CH4 profiles from ACE-FTS where the precision and accuracy is more than an order of magnitude worse than the TCCON standards for column retrievals. Instead, they could have done an extensive comparison with data from the Sodankylä site but for an extended time period. And there are more TCCON sites that have been covered by aircore measurements than just Sodankylä. One is even part of this study: Orleans!

In the troposphere, it is even more puzzling why the bulk of available in-situ aircraft profile data was completely ignored. Even during the same time period, several TCCON sites were overflown by the NASA ATom campaign. And several of the sites that have been used in this study had aircraft profiles taken during the IMECC campaign in 2009. Why did they not process spectra from that time period for these sites instead? There is no obvious reason why they limited themselves to 2016-17. Given these issues, I cannot recommend publication without major revisions. The authors should try their profile retrieval on stations and time periods with corresponding aircraft or aircore in-situ CH4 profiles. If they do so, the whole section 3.3 (ACE-FTS comparison) could be dropped. The section 3.4 (TCCON proxy) would still be useful to evaluate if the proxy method or a profile retrieval provides better results.
General comments:

Why were these sites selected? Among those, Sodankylä is the only one where in-situ profiles were used to validate the retrieval. Others sites do have profiles but they were ignored. To my knowledge, two sites never had any kind of in-situ profiling but were included nonetheless.

Aircraft/aircore profiles at the selected sites:

- Sodankylä: many during several campaigns (but only a limited set used here)
- Bialystok, Bremen, Orléans: aircraft profiles taken 2009 but not used here. Data published in Geibel et al.: Calibration of column-averaged CH4 over European TCCON FTS sites with airborne in-situ measurements, Atmos. Chem. Phys., 12, 8763-8775, https://doi.org/10.5194/acp-12-8763-2012, 2012.
- Ny-Alesund: never
- St Denis: never

ATom visited several TCCON sites in 2016/17, the data is publicly available. Why was this data set completely ignored?

Why was the time frame 2016-17 selected? There is no special reason given and I cannot see one that requires to try the SFIT4TCCON on the same time period for all sites. It would make much more sense to use it on persiods with available in situ profiles for each site. Why was St. Denis used as the prototype? It is probably the wettest site of all and not necessarily characteristic for the others.

One of the main concerns expressed by the referee is the reason behind the chosen time period (2016-2017) and the chosen sites. This paper is the result of work done in the EU RINGO project in which one work package was dedicated to demonstrate the feasibility of vertical profiles retrievals of CH4 from TCCON spectra. The study is based on six TCCON sites from the RINGO partners in the project: BIRA, University of Bremen and FMI. The initial setup of the RINGO project and the practical limitations were the main reasons to limit this study to the analysis of 2 years of data at these six sites. We agree with the referee and are willing to expand this study to all TCCON sites but this will cause significant delay: firstly an agreement between BIRA and all TCCON PIs is necessary and secondly additional resources to cover IT costs need to be found (and BIRA has no perspective on this in the near future). We also believe that the six chosen sites are representative for the majority of the TCCON network covering a large latitude band. To extend the SFIT4TCCON retrievals to more TCCON stations, we need to get the spectra from the individual TCCON PIs. We would like to mention that this work was presented to the entire TCCON community at the IRWG-TCCON annual meeting in May this year at Wanaka, New Zealand. We invited the TCCON community to take part in this study, given the technical limitations we are facing. However, until now, only the spectra from these six sites are available. In this paper, we present the CH4 profile retrieval method, and we can make the CH4 profile retrieval easily for other time period and at other sites when the NIR spectra and extra supports are available in the future.

Concerning the in-situ comparisons: among the six sites, Bialystok, Bremen and Orleans have been validated in the past with IMECC aircraft measurements (Geibel et al., 2012) and Sodankyla has been calibrated with AirCore measurements. As the referee mentions, the Sodankyla AirCore data have been used in this study to validate the vertical profile retrieval results and we will add to the revised version a comparison against all available IMECC aircraft profiles for the Bialystok, Bremen and Orleans sites (not necessarily within the chosen time period). The AirCore measurements at Orleans are not added in the revised version, and the reason is discussed later. Unfortunately, no Atom profile (https://acd-ext.gsfc.nasa.gov/Missions/ATom/Flights/) is available at these six sites.

St. Denis is used as the prototype, as the site is very humid and the H2O is very important in the CH4 retrieval (as we can see the retrievals from Band 1 have an artificial curve): if the retrieval method works in humid conditions, then also in dry conditions.

Specific comments:
The data product is called SFIT4TCCON but - unlike the name suggests - it is not an official TCCON data product. Was the use of the name "TCCON" approved by the TCCON PIs? I am a TCCON PI myself but I am not aware of this.
Thanks for the suggestions. We agree that the SFIT4TCCON is not appropriate. So we use "SFIT4NIR" as the new name for our retrievals.

Section 2.2.1:
- p. 4, l. 11: Diurnal variations above St Denis are assumed to be small. What about the biomass burning season, would this still be true? At least there should not have been much biomass burning activity during the chosen time frame. It would have been easy to check other sites TCCON sites for these criteria. Also, I do not understand why the band with the highest correlation between CH4 and dry air pressure is best choice.
Thanks for the suggestions.
We realize that it is not appropriate to use the correlation between the dry air column and the CH4 column to select the retrieval window. The dry air total column and CH4 total column are correlated if we assume that the XCH4 is relatively constant during the day. There is no strong CH4 emission at St Denis, especially in non-biomass burning period. However due to the vertical and horizontal transport, and the reaction with OH, the XCH4 can be variable. According to the TCCON measurements, the daily standard deviation of XCH4 is within 3 ppb (~0.16%), while the daily standard deviation of dry air total column is about 0.14%. As the two standard deviations are in the same order, it is very difficult to use the variation of the dry air column to select the variation of the CH4 total column. We think this is a very interested topic but might beyond the scope of this paper. In order to avoid the confusion, in the revised version, we will remove this paragraph and the argument to select band 2 will be based on the fact that the fit in Band 2 is the best and less affected by $H_2O$.

- p. 6, l. 8-10: the "Sa^-1 = ..." should be a numbered equation. The quantity "L1" is not described at all, "T" only very briefly. Provide a reference linking Tikhonov 1963 to the Sa matrix. In the cited reference, Tikhonov certainly did not refer to Rodger's original work which came years later (1976).
This will be taken into account in the revised version.

Section 2.2.4:
- How was the SFIT4 column AVK calculated?
The SFIT4 implementation uses optimal estimation and the methane VMR profile is in the state vector in relative units (relative to the CH4 apriori). The column averaging kernel (CAVK) can be calculated from this 2D state vector averaging kernel (calculated analytically) using standard methods.

Section 2.2.5:
- 5% uncertainty might be fine for St Denis sources but may underestimate the variability closer to CH4 sources. Check available aircraft profiles (e.g. from ATom, HIPPO etc.) as well as aircore to get a better grip on the expected variability. The model data might be too smooth.

Thanks for the recommendation. Unfortunately, there are no CH4 in situ profiles (aircraft or AirCore) available at St Denis, and even at Sodankyla, we only have a few AirCore measurements. Therefore, in this study, the random a priori covariance matrix is estimated from the covariance matrix of the WACCM 492 monthly means (41 years). For the systematic uncertainty, we have chosen a value of 5% (about 90 ppb in the troposphere), based on the difference between the SFIT4TCCON a priori CH4 mole fraction near the surface and the local in situ measurements (Zhou et al., 2018). As CH4 is relatively stable in the atmosphere with a life time of 9-10 years, it is assumed that 5% systematic uncertainty is acceptable for all altitudes. This is also only relevant for the smoothing uncertainty which is cancelled in the comparison with the in situ profiles since we apply the smoothing equation.

Eq. 7: Why did you use pressure and not the O2 column for the calculation dry air column? There are good reasons why TCCON abandoned surface pressure as a proxy for airmass years ago.

By using the ratio between the target species and O2 can reduce the uncertainties common to both gases (Yang et al., 2002), e.g. the surface pressure and solar tracker pointing. However, there is a systematic uncertainty in the spectroscopy of O2, as the Xair of TCCON data is close to 0.98. In addition, the retrieved O2 is also affected by the airmass, so that the TCCON retrievals are corrected with an airmass dependent factor. For the CH4 profile retrieval, the O2 retrieval is not simultaneously performed, and we do not apply the airmass dependent and independent correction for our retrieval. The uncertainty of the surface pressure at these TCCON sites is within 0.1 hPa, which means that the uncertainty of the dry air total column derived from the surface pressure is within 0.1%. By comparing the XCH4 between the TCCON standard retrievals and SFIT4NIR retrievals, the bias of the SFIT4NIR retrievals are within the TCCON reported uncertainty.

Section 3.2
- It is doubtful that surface in-situ measurements from 20 km away are representative for St Denis. If you think otherwise, you should provide some reasonable arguments.

We think comparing the tropospheric XCH4 at St Denis and in situ measurement at Maido is reasonable. Some arguments are added in the revised version.
CH4 is well-mixed in the lower atmosphere with a life time of 8–10 years (Kirschke et al., 2013). Maido is located in the high mountain (2155 m), which is less affected by the surface. In addition, there is no strong methane emission nearby.

Section 3.3
- I understand the need to find something to compare the stratospheric profiles with. However, given the very limited precision and accuracy listed here for the 10-year old ACE-FTS data (25%), I wonder how useful this comparison can be. One might be better off comparing to zonal means with better statistics. That might also provide profiles for the stations where reasonable co-locations could not be found.

In the revised version: "The older version v2.2 data of the ACE-FTS CH4 data have been compared to space-based satellite, balloon-borne and ground-based FTIR data (De Maziere et al., 2008). The accuracy of the version 2.2 data is within 10% in the upper troposphere - lower stratosphere (UTLS), and within 25% in the middle and higher stratosphere up to the lower mesosphere. The uncertainty of the new version of the ACE-FTS data has a reduction of about 10% near 35-40 km and a slight reduction at 23 km (Waymark et al., 2014). ".
Therefore, the uncertainty of the stratospheric XCH4 of each individual ACE-FTS measurement should be around or within 10%. As we select the ACE-FTS measurements

within the ±3x30 (latitude x longitude) co-located box around each site, normally there are more than 1 ACE-FTS measurements. At Orleans and Bialystok, the mean co-located number of ACE-FTS measurements for each FTIR measurement day is about 4.0. At St Denis, the mean co-located number of ACE-FTS measurements for each FTIR measurement day is about 3.2. The mean of the co-located ACE-FTS measurements is applied to compare with the FTIR measurement. By doing the average, the uncertainty of the ACE-FTS measurement is reduced to about 5% (assuming all random uncertainty, then $\sqrt{N} \approx 2$). Figure 8 in the revised version shows that the seasonal cycles of the stratospheric XCH4 from the ACE-FTS and SFIT4NIR measurements are close to each other. The amplitude of the seasonal cycle of the stratospheric XCH4 is about 15%, which is larger than the uncertainty of the ACE-FTS measurements.

- If you decide to make the major revisions that I recommend, this whole section could be dropped.
We prefer to leave this section because it can show the seasonal cycle of the stratospheric XCH4.

Section 3.4
- Due to the H2O interference in the HF microwindow, I would trust the N2O method more for a tropical site like St. Denis.
Yes. The HF absorption line at a humid site is strongly affected by the H2O line.

- p. 14, l. 10 to p. 15, l. 4: it should not be too difficult to find out which of the raised possibilities 1) to 3) are true. The source code for the TCCON retrieval as well as the handling of the a priori profiles is available.
The possible reasons are mainly from the proxy method, which are from its assumption. Further investigation should be carried out to figure out which one is the dominated reason.

Section 3.5
- I believe that the Aircore measurements are the most trustworthy validation source for the tropospheric and stratospheric profiles. There are other TCCON sites which had simultaneous aircore profiles taken. Why were these not used? This would also improve the statistics in Fig. 11, which suffers from the fact that a very limited range of XCH4 was compared.
Thanks for the suggestion. We add the comparison between the IMECC aircraft measurements and SFIT4NIR retrievals in the revised version (Section 3.6). The SFIT4NIR retrievals are carried out on these aircraft overflight days. Since the aircraft profile only covers the vertical range from ~300 to 13000 m, we only compare their tropospheric parts. The aircraft measurements confirm that there is an about 1.0% overestimation in the SFIT4NIR tropospheric XCH4, which is consistent with the result from the comparison between the AirCore measurements and SFIT4NIR retrievals.

[Figure]

Figure 1. The scatter plots of XCH4 between the SFIT4NIR and the IMECC aircraft measurements together with the AirCore measurements for the tropospheric components. The black line is the one-to-one line and the dashed red line is the regression line with the intercept to zero (y = a · x). N is the co-located measurement number, R is the correlation coefficient, and a is the slope.

The AirCore measurements at Orleans are not used in the paper, because there is an underestimation of 0.5% in the AirCore XCH4 measurements. Figure 2 shows that the AirCore XCH4 is 0.5±0.2% lower than the SFIT4NIR retrievals, and the AirCore XCH4 is 0.5±0.2% lower than the TCCON retrievals. Note that, the AirCore data has been smoothed with SFIT4NIR/TCCON AVK, when comparing with SFIT4NIR/TCCON retrievals. The SFIT4NIR retrieved XCH4 is close to the standard TCCON retrieval, which is consistent with the results in Section 3.1. As an example, the time series of the XCH4 from the AirCore, TCCON and SFIT4NIR measurements on 17 February 2017, together with their profiles and differences are shown in Figure 3.

If we use the AirCore measurements at Orleans to validate the SFIT4NIR tropospheric and stratospheric XCH4, there is an overestimation of 1.0±0.5% in the SFIT4NIR tropospheric XCH4, and an underestimation of 1.7±0.7% in the SFIT4NIR stratospheric XCH4. Since the result in the troposphere is in good agreement with the results from the AirCore at Sondakyla and IMECC aircraft measurements at Orleans, Bremen and Bialystok, it is probably that the stratospheric part of the AirCore measurement at Orleans is underestimated. There is one study from the Bremen group, they also found that the AirCore CH4 measurements at Orleans is about 0.5% larger than the TCCON measurements (after removing the two outliers) https://tccon-wiki.caltech.edu/@api/deki/files/3129/=TCCON_2019_Petri.pdf (Page 8; please use your TCCON wiki account to login).

It is not clear to us why there is an underestimation in the AirCore data, partially in the stratosphere. The AirCore measurements (LSCE/LMD) in 2016 and 2017 at Orleans used here have not been compared with other in situ profiles. In June 2019, there is an AirCore iner-comparison campaign within the RINGO project of 12 AirCore systems from 7 research team (LSCE/LMD, NOAA/ESRL, University of Groningen, Goethe University Frankfurt, University of Bern, Finnish Meteorological Institute, and Forschungszentrum Jülich) and of 1 sampling balloon system (https://www.icos-ri.eu/ringo/news-and-events). Within this campaign, we have the opportunity to understand better the uncertainty of the AirCore measurements. Note that the AirCore measurements (RUG) in 2016 and 2017 at Sodankyla has already been

compared with NOAA and GUF ones, so that we understand the uncertainty of the Sodankyla AirCore measurements better.

In conclusion, we do not include the AirCore measurements at Orleans in this paper, and further investigation is required before using the AirCore CH4 measurements at Orleans.

[Figure]

(Page 8 in Petri et al., 2019, TCCON annual meeting)

In case that the reader does not have the access to the TCCON wiki, we attach the slide 8 here.

[Figure]

Figure 2. Left panel: the scatter plot between the smoothed AirCore and SFIT4NIR XCH4 daily means. Right panel: the scatter plot between the smoothed AirCore and TCCON XCH4.

[Figure]

Figure 3. Upper panel: the time series of the XCH4 from the AirCore, TCCON and SFIT4NIR measurements on 17 February 2017. Lower panels: the CH4 profiles from AirCore and SFIT4NIR measurements (left) with their differences (right).

[Figure]

Figure 4. Left panel: the scatter plot between the smoothed AirCore and SFIT4NIR tropospheric XCH4 daily means at Orleans. Right panel: the scatter plot between the smoothed AirCore and SFIT4NIR stratospheric XCH4 daily means at Orleans.

References:

- All the TCCON dataset citations are wrong: they still refer to Oak Ridge Naional Laboratory from which the TCCON data archive moved away already in October 2017! Strange how this can happen when several TCCON PIs are listed as coauthors.
Corrected

- Acknowledgments: the authors should check if the acknolwedgments are in line with what is required by the TCCON Data Use Policy.
Checked

Minor comments:
- p. 3, l. 15: I would not call that "noise".
- p. 3, l. 21: Please explain the ATM acronym at least once.
- Fig. 2: poor color choice for dry air pressure as green is already used for band 3.
- p. 6, l. 5: should be "constrain" instead of "constraint"
- p. 7, l. 5: should be "panel" instead of "penal"
- p .8, l. 17: "The other retrieved parameters have do not contribute ..."
- p. 8, l. 28: "retrieved CH 4 total column are 3.2 and 0.5%" units missing
- Table 5: Why not just write "
Corrected

Reference:

Geibel et al.: Calibration of column-averaged CH4 over European TCCON FTS sites with airborne in-situ measurements, Atmos. Chem. Phys., 12, 8763-8775, https://doi.org/10.5194/acp-12-8763-2012, 2012.

Kirschke, S., Bousquet, P., Ciais, P., Saunois, M., Canadell, J. G., Dlugokencky, E. J., Bergamaschi, P., Bergmann, D., Blake, D. R., Bruhwiler, L., Cameron-Smith, P., Castaldi, S., Chevallier, F., Feng, L., Fraser, A., Heimann, M., Hodson, E. L., Houweling, S., Josse, B., Fraser, P. J., Krummel, P. B., Lamarque, J.-F., Langenfelds, R. L., Le Quéré, C., Naik, V., O'Doherty, S., Palmer, P. I., Pison, I., Plummer, D., Poulter, B., Prinn, R. G., Rigby, M., Ringeval, B., Santini, M., Schmidt, M., Shindell, D. T., Simp- son, I. J., Spahni, R., Steele, L. P., Strode, S. A., Sudo, K., Szopa, S., van der Werf, G. R., Voulgarakis, A., van Weele, M., Weiss, R. F., Williams, J. E., and Zeng, G.: Three decades of global methane sources and sinks, Nat. Geosci., 6, 813–823, https://doi.org/10.1038/ngeo1955, 2013

Yang, Z., Toon, G. C., Margolis, J. S., and Wennberg, P. O.: Atmospheric CO2 retrieved from ground-based near IR solar spectra, Geophys. Res. Lett., 29, 53–1–53–4, https://doi.org/10.1029/2001GL014537, 2002.

Zhou, M., Langerock, B., Vigouroux, C., Sha, M. K., Ramonet, M., Delmotte, M., Mahieu, E., Bader, W., Hermans, C., Kumps, N., Met- zger, J.-M., Duflot, V., Wang, Z., Palm, M., and De Mazière, M.: Atmospheric CO and CH4 time series and seasonal variations on Reunion Island from ground-based in-situ and FTIR (NDACC and TCCON) measurements, Atmos. Chem. Phys., 18, 13 881–13 901, https://doi.org/10.5194/acp-18-13881-2018, 2018.

---

## Author Comment (AC1)

*Black: referee's comments red: authors' answers*
*First of all, we want to thank the two referees for the detailed analysis of our paper.*
*For the details, please look into the paper with keeping track of changes.*

Anonymous Referee #2

The study by Zhou et al. employs the full-physics retrieval code SFIT4, which is used by the Network for the Detection of Atmospheric Composition Change (NDACC) in order to retrieve vertical profile information on atmospheric methane from solar absorption spectra measured in the near infrared (NIR) by spectrometers within the Total Carbon Column Observing Network (TCCON). Comparisons of retrieval codes lead to improvements in the codes and therefore, this study is a contribution to remote sensing measurements of atmospheric CH4. I recommend its publication in AMT after the questions, issues and comments outlined below have been addressed.

Major Comments:
The authors state that the ILS parameters are retrieved simultaneously by the code. How does the retrieved instrument line shape look like and how constant is it for all the sites involved? The Bruker 125HR spectrometers exhibit excellent ILS stability, so the retrieved values should reflect this. Therefore, it would be beneficial if the authors could show a time series of the ILS and the parameters.

Thanks for the suggestions. We plot the time series of the retrieved amplitude error of the modulation efficiency at the maximum optimal path difference (MOPD) at the six sites (see Figure 1). As expected, the retrieved ILS are very constant, and the retrieved amplitude errors are within 2%. In the revised version, more information of the retrieved ILS is added in the retrieval strategy Section.

[Figure]

Figure 1. the time series of the retrieved amplitude of the modulation efficiency at maximum path difference (45 cm) at six sites.

The profile retrieval relies on the Alpha values, as discussed in Sec 2.2.3, but could the authors please explain the physical significance of the Alpha value?

The physical significance of the Alpha value is the correlation among layers. With a larger Alpha value, the stronger relationship among layers are constrained. Some explanations are added in the revised version.

Also, it seems to me, as shown in Fig. 3, that the retrieved profile just approaches a scaled value of the a-priori profile at Alpha values of 1,000 and 10,000. Optional addition to Fig 3: Could the authors add a plot of retrieved VMR profile divided by the a-priori VMR profile with altitude as y-axis or something similar? This is to show how much the a-priori is scaled and the altitude dependence of this value.

Following the referee's suggestion, we add one ratio profile in the revised version, along with the original plot.

In TCCON, the Xair value and its time series are indicative of instrument stability, I think a comparison of the SFIT retrieved Xair and the TCCON Xair for the sites is warranted for this study.

The Xair in TCCON data is defined as: Xair = $VC_{dryair}/(VC_{O2}/0.2095)$, where $VC_{dryair}$ is calculated from the surface pressure and total column of water vapor. The Xair is often used by TCCON data user as an index to check the time error and instrument stability. However, in SFIT4TCCON retrieval, we do not retrieval $O_2$ and the dry air column is derived from the surface pressure and total column of water vapor directly. Therefore, there is no Xair for SFIT4TCCON data.

In situ measurements: In its current state, I do not see the full usefulness of the comparison between the in situ ground-based measurements and the tropospheric product of SFIT4TCCON (Sec 3.2). Both measurements have completely different sensitivities, as the authors mentioned, and I think comparing the time-series alone does not sufficiently provide information to say that "The SFIT4TCCON tropospheric and stratospheric XCH4 can observe the CH4 seasonal variation very well, which has been confirmed by the ground-based in situ measurements. . ." in the conclusions. For example, the agreement between SFIT4TCCON tropospheric CH4 and in situ looks to be closer during the winter months and farther during the summer months both at Orleans and St. Denis. But it is difficult to see from the scattered, overlapping data points. I recommend that the authors derive a seasonal fit and/or trend line to both FTS and in situ data. Alternatively, a correlation/scatter plot for the FTS vs. in situ would be useful, like in Fig 6.

Thanks for the suggestion. We now compare the monthly means from the in situ measurements and tropospheric product of SFIT4TCCON at Orleans and St Denis. The seasonal cycles from two measurements have similar phases and amplitudes, which can support our conclusion.

Finally, it wasn't clear to me until the last sentence of Sec 2.2.1 that the SFIT4TCCON retrieval does not actually use all the CH4 bands used by TCCON, so the term "SFIT4TCCON" is potentially misleading to the readers and data users. I recommend that this term be changed. Moreover, the authors state in the abstract that "the SFIT4 retrieval code is applied to retrieve CH4 mole fraction vertical profile using TCCON spectra". First, this is not entirely true because the retrieval only uses a subset of the NIR spectra used for retrieval of CH4 in the TCCON network. Second, the term "TCCON spectra" is a term that is not officially used by TCCON and this is also not a standard TCCON product, so I do not think that this term is appropriate to be used to describe the spectra used in this study.

Thanks for the suggestions. We agree with you that the SFIT4TCCON is not appropriate. So we use "SFIT4NIR" as the new name for our retrievals.

Technical/Minor Comments:
P1, Line 7: two distinct species -> two distinct pieces
Corrected

P1, Line 19: spectrometer -> spectrometers
Corrected

P2, Line 16: the atmosphere chemistry -> atmospheric chemistry
Corrected

P2, Line 23: started to increasing -> started to increase
Corrected

P2, Line 24: remove "the" in: partly by the getting -> partly by getting
Corrected

P3, Line 15: It seems that the spectra is converted to SFIT4 readable format then corrected for SIV, please arrange sentences if not the case.
Corrected

Figure 2. How do the retrieved columns for each band look like for the standard TCCON product? Does the TCCON CH4 at band 1 also exhibit the same curve?
We checked the TCCON CH4 (GGG2014) retrieved total columns from the three bands, and no such large curve is observed. However, the retrievals from Band 1 also show opposite diurnal variation compared to the results from Band 2 and 3.

[Figure]

Figure 2. the retrieved CH4 total columns from three bands in the GGG2014 code at St Denis on 30 July 2016.

Although the TCCON ATM spectroscopy is used in our retrieval (SFIT4NIR), we notice that the spectrum in Band1 is not well fitted with SFIT4 code. More investigations are needed in the future to understand the reason. So far, we only choose the Band2 to do the CH4 profile retrieval.

P6, Line 3: The DOFS definition is not entirely accurate; moreover it is not consistent with line 7 of the Abstract.
Corrected. According to Rodgers (2000) Page 30, the DOFS is defined as "degree of freedom for signal".

P6, Line 5: The sentence about S_(epsilon) needs to be checked, it seems wrong. Corrected

P6, Line 5: change "to constraint" to "to constrain" Corrected

P6, Line 5: "to determine whether the" or "to determine if the" Corrected

P7, Line 5: change "penal" to "panel" Corrected

P8, Line 17: remove "have" in "parameters have do not" Corrected

P8, Line 19: change "error" to "errors" Corrected

P10, Lines 25 and 26: use plural "measurements .... are "Corrected

P10, Line 28: Either quantify how well they are calibrated or change "well calibrated" to just "calibrated"
Corrected

Fig. 5: The panels are too crowded and the legend boxes partially cover the data. I think this should be improved. The TCCON and SFIT4TCCON data overlap and it is impossible to determine the data points. This figure needs to be revised. The same goes for Fig. 7. Fig. 7, Legend: change "Insitu" to "in situ". Title: fix "stdenis" and "orleans"
Corrected. Note that the number of the co-located hour means are slightly changed, because the official TCCON data have been updated at Caltech website.

Fig 8: The legends overlap with the actual data points, making it hard to read the figure. Additionally, Fig. 8 could be improved by adding a correlation plot for each panel because the data are too sparse and does not cover a long time series. In fact, the correlation plots could be a better representation.
The correlation plots are added.

P13, Line 9: There have been other validation activities after De Maziere et al., 2008 and the results for CH4 have improved since then, e.g. https://doi.org/10.4401/ag-6339
Added in the revised version.

P13, Line 19: "sits" to "sites" Corrected

Figure 9: I would like to know how the data points in Fig. 9 are treated. Daily means, hourly means? How is the filtering done and how are the errors in each retrieval taken into account? The answer to this could explain or support the statement "at St Denis (a moist site), the TCCON HF retrievals are strongly affected by H2O so that TCCON proxy method tropospheric and the stratospheric XCH4 data using HF have many outliers"
These are hourly means. Added in the revised version.

P13, Line 26-27: This "slight seasonal and site dependent bias" is not clear to me from the figure.
One sentence is added in the revised version.
"For example, the difference in tropospheric $X_{CH_4}$ between $N_2O$ proxy method and HF proxy method is larger in summer than that in winter at Ny-\r{A}lesund. "

P14, Line 3: "systematic larger" -> "systematically larger"
Corrected

P14, Line 5: The sentence starting from line 5 and ending in line 6 needs to be improved.
Corrected

Fig. 10 Caption: it is very hard to see the "scaled SFIT4TCCON a priori profile (dotted black line)"
Adapted.

Fig. 10 and Sec 3.5: Why and how is the AirCore profile smoothed? It seems that there is a lot of structure and information in the AirCore profile that is lost from the smoothing. Also, the sentence "The extended" AirCore profile is then smoothed with the closest SFIT4TCCON retrieval" is not very clear.
According to Rodgers et al., (2003), the in situ vertical profile with a fine vertical resolution must be smoothed to compare with the optimal estimation method retrieval from the remote sensing technique with a coarse vertical resolution. Therefore, the AirCore profile is smoothed to consider the vertical sensitivity of the FTIR retrievals. The method to do the smoothing is added in the revised version.

The plot is adapted.

Fig. 11: The authors need to provide an error estimate on the slopes of the lines.
Added

P18, Lines 13-15: I think measurements from a single tower compared to a single TCCON site are not sufficient to arrive at this conclusion. Moreover, the authors should quantify what they mean by "very close".
Thanks. To avoid the confusion, the sentence is removed.

P18, lines 17-20. These sentences seem contradicting. On one hand the authors state that "there is almost no systematic bias between the SFIT4TCCON and AirCore XCH4", but on the other hand the next sentence state "An overestimation of 1.2% in the SFIT4TCCON tropospheric XCH4 and an underestimation of 4.0% in the SFIT4TCCON stratospheric XCH4 is seen by comparing with AirCore measurements"
Adapted in the revised version.

Reference:
Rodgers, C. D.: Intercomparison of remote sounding instruments, J. Geophys. Res., 108, 46–48, https://doi.org/10.1029/2002JD002299, 2003.